# Non-redundant roles for the human mRNA decapping cofactor paralogs DCP1a and DCP1b

Ivana Vukovic[1] , Samantha M Barnada[1] , Jonathan W Ruffin, Jon Karlin[1], Ravi Kumar Lokareddy[2], Gino Cingolani[2], Steven B McMahon[1]

Eukaryotic gene expression is regulated at the transcriptional and post-transcriptional levels, with disruption of regulation contributing significantly to human diseases. The 5′ m7G mRNA cap is a central node in post-transcriptional regulation, participating in both mRNA stabilization and translation efficiency. In mammals, DCP1a and DCP1b are paralogous cofactor proteins of the mRNA cap hydrolase DCP2. As lower eukaryotes have a single DCP1 cofactor, the functional advantages gained by this evolutionary divergence remain unclear. We report the first functional dissection of DCP1a and DCP1b, demonstrating that they are non-redundant cofactors of DCP2 with unique roles in decapping complex integrity and specificity. DCP1a is essential for decapping complex assembly and interactions between the decapping complex and mRNA cap-binding proteins. DCP1b is essential for decapping complex interactions with protein degradation and translational machinery. DCP1a and DCP1b impact the turnover of distinct mRNAs. The observation that different ontological groups of mRNA molecules are regulated by DCP1a and DCP1b, along with their non-redundant roles in decapping complex integrity, provides the first evidence that these paralogs have qualitatively distinct functions.

## Introduction

mRNAs represent a distinct functional RNA group whose regulation is a critical aspect of eukaryotic gene expression. Most mature mRNAs in eukaryotic cells are capped by 7-methylguanosine at their 5′ terminus (m[7]G) (1, 2, 3). The mRNA cap controls critical stages of the mRNA lifecycle: recruiting complexes involved in mRNA processing, mRNA export, translation initiation, protecting mRNA from degradation by 5′-3′ exoribonucleases, and marking mRNA as "self" to avoid being targeted by the innate immune system (1, 4). Because the mRNA cap plays these essential roles in the mRNA lifecycle,

precise regulation of mRNA capping and decapping is critical (1, 2, 5).

mRNA capping occurs co-transcriptionally through three distinct enzymatic reactions (5, 6, 7). RNA triphosphatase first removes the 5′ γ-phosphate from the nascent pre-mRNA to generate a 5′ diphosphate end (6, 7, 8). RNA guanylyl transferases then cap the pre-mRNA with GMP (6, 7, 8). Finally, RNA guanine-N7 methyltransferases transfer a methyl group from the donor S-adenosylmethionine to the N7 position of the terminal guanine base (6, 9). Cap-binding complex binds to the mRNA cap and recruits proteins that facilitate further mRNA processing (e.g., splicing and polyadenylation) before the mRNA is exported to the cytoplasm (1, 10, 11). The cap-binding proteins protect mRNA from degradation by physically blocking the cap from being recognized and hydrolyzed by the decapping enzymes (12). In the cytoplasm, mature mRNA has several different fates: (1) translation into protein, (2) holding in processing bodies (PBs) in a translationally repressed state, or (3) degradation (13). The binding of the eIF4F complex, consisting of the initiation factors eIF4E, eIF4G, and helicase DDX6, to the mRNA cap initiates mRNA translation (10, 14, 15). PBs are cytoplasmic non-membranous ribonucleoprotein granules that contain high concentrations of RNA and RNA-binding proteins (16, 17, 18). Decapping complex members and exoribonucleases are among the proteins localized to PBs, leading to a model in which PBs are also a site of mRNA decapping and subsequent degradation (16, 18, 19). More recently, it has been discovered that PBs store translationally repressed mRNAs (19, 20). This translational repression of mRNA associated with decapping is reversible, providing another method of gene regulation (19). Similarly intriguing is the role of transcript buffering (21, 22, 23, 24). Transcript buffering was first described in *Saccharomyces cerevisiae*, where inhibition of mRNA transcription or mRNA decay pathways activates a buffering system where mRNA abundance is maintained at a constant level through reciprocal adjustments in transcription initiation and mRNA decay. However, the mechanism behind these observations is not well understood (22).

As with all aspects of mRNA metabolism, the degradation of mRNA is tightly regulated (25, 26). Whereas the bulk of cellular mRNA is degraded through decapping, followed by 5′-3′

---

[1]Department of Biochemistry and Molecular Biology, Thomas Jefferson University, Philadelphia, PA, USA [2]Academic Joint Departments - Biochemistry & Molecular Genetic, University of Alabama at Birmingham, Birmingham, AL, USA

Correspondence: Steven.McMahon@jefferson.edu

degradation mediated by the exoribonuclease XRN1, some mRNAs are degraded by the exosome and a 3′-5′ degradation pathway (4, 25, 27). Degradation is initiated by the shortening of the polyA tail by deadenylases (13, 28). Human cells contain a family of NUDIX proteins, several of which have intrinsic mRNA decapping activity in vitro. Among the NUDIX proteins which hydrolyze the m$^7$G mRNA cap are DCP2/Nudt20 and Nudt16 (4, 29). The N-terminus of human DCP2 contains a regulatory domain (RD), followed by a catalytic domain (CD) (30), whereas the C-terminus is intrinsically disordered (31). The CD consists of a conserved NUDIX hydrolase domain which hydrolyzes diphosphates linked to nucleosides (30, 31). The CD and RD domains of DCP2 are linked by a flexible hinge and undergo a rapid transition between an open and closed state (30, 31, 32). Because of its flexible nature, defining the structure of the active and inactive conformations of DCP2 has been challenging (30, 31, 33). Most known structures of DCP2 are based on the *Schizosaccharomyces pombe* or *Kluyveromyces lactis* proteins (31). DCP2 has low basal decapping activity in vitro, and its enzymatic activity is greatly enhanced by its interaction with its main decapping activator, DCP1 (31, 34). DCP1 contains an Ena/Vasp homology domain 1 (EVH1 domain), known to mediate interactions of multiprotein complexes and to transduce migratory and morphological signals into the cytoskeleton, at its N-terminus. DCP1 proteins also contain a trimerization domain (TD) at their C-terminus (35). DCP1 binds to DCP2 through an asparagine–arginine–containing *loop* (NR loop) in the EVH1 domain, an interaction that converts the DCP2 conformation from an inactive/open to an active/closed state (30). As mentioned above, the human genome encodes two DCP1 paralogs, termed DCP1a and DCP1b. DCP1a and DCP1b can each homo- and hetero-trimerize via their TD, and it is the trimeric form of DCP1a/DCP1b that is found with DCP2 in the decapping complex (36). In humans, the interaction between DCP2 and DCP1 is weak and is stabilized by the binding of the decapping enhancer protein EDC4 (30). From a functional perspective, efficient in vivo decapping activity requires the binding of EDC4 to DCP1 and DCP2, with EDC4 depletion leading to decreased decapping (30).

To date, mechanistic studies have focused primarily on the single DCP1 protein in yeast and the DCP1a paralog in humans. Currently, we lack an understanding of the overlapping and unique aspects of DCP1a and DCP1b function in humans. The studies reported here broaden our general understanding of mRNA decapping by demonstrating that DCP1a and DCP1b indeed have unique roles in human cells. We report that whereas DCP1a is critical for decapping complex integrity, both DCP1a and DCP1b foster unique interactions between decapping complex subunits. The decapping complex exists in multiple configurations, and stoichiometry of DCP1a and DCP1b in the complex varies in these distinct configurations. Furthermore, DCP1a and DCP1b impact the stability and transcription of distinct sets of mRNAs. For example, DCP1a controls mRNAs that encode proteins with a role in adaptive immunity and transcription, whereas a separate group of DCP1b-dependent mRNAs also play a role in transcription. These findings provide the first functional dissection of the distinct roles of the human DCP1 paralogs in mRNA metabolism.

# Results

## DCP1a, and not DCP1b, mediates the interaction of DCP2 with EDC4 and EDC3

Whereas early eukaryotes have a single DCP1 gene, the human genome has evolved to encode two paralogs, DCP1a and DCP1b. One advantage of this divergence could be tissue-specific roles and expression patterns for the two paralogs. However, expression analyses across different tissues in humans revealed that DCP1a and DCP1b exhibit concordant expression in some tissues (e.g., cerebral cortex, thyroid gland, spleen) and discordant expression in other tissues (e.g., bone marrow, colon) (Fig S1). This raises the possibility that DCP1a and DCP1b have evolved to harbor distinct, non-redundant functions in the tissues where they are co-expressed. This is supported by the fact that DCP1a and DCP1b appear in the decapping complex as heterotrimers, potentially allowing decapping complex stoichiometry to be tailored for context-specific needs.

An initial functional distinction between the paralogs is that the DCP1b promoter is tightly bound by the stress-induced transcription factor p53, whereas the DCP1a promoter is not (37) (Fig S2A). This binding suggests the possibility that DCP1b is transcriptionally regulated by p53, whereas DCP1a is not. To directly assess the contribution of the high-affinity p53 binding site in the DCP1b promoter, the p53 consensus response element was mutated via CRIPSR-Cas9 in a WT p53 human cell line (HCT116; Supplemental Materials). Upon p53 activation with the MDM2 inhibitor Nutlin-3a, or the genotoxic chemotherapy agent 5FU, transcription of DCP1b was up-regulated in parental cells but not in the clones carrying the mutant p53 response element (Fig S2B). The level of induction of DCP1b is similar to that of established p53 target genes, such as BAX (Fig S2C).

Understanding any divergent roles for DCP1a and DCP1b requires an understanding of their respective interactions with other components of the mRNA decapping machinery. As mentioned, yeast has a single DCP1 ortholog. However, residues that mediate the DCP1 and DCP2 interaction in yeast are not conserved in metazoan cells (33). The web resource tool was used as described to predict the low-complexity regions in the human and yeast versions of DCP1 and DCP2. These regions are expected to lack tertiary structure. Yeast DCP2 has a long, disordered C-terminal tail that contains eight low-complexity regions which themselves contain short linear motifs (Fig 1A) (38). DCP2 interacts with other decapping complex members in large part through the avidity effects of short linear motifs (38). The C-terminal tail of human DCP2 is shorter than that of its yeast counterpart, and several of the low-complexity regions have been transferred to other subunits of the decapping complex, specifically the enhancers of decapping (EDC) proteins and DCP1a/b (38) (Fig 1A). Both DCP1a and DCP1b also contain low-complexity motifs (Fig 1A), suggesting that the human decapping complex has undergone significant evolutionary re-wiring (38). As such, mechanisms mediating assembly and integrity of the yeast complex provide little information relevant to the human complex. As a key example, it is unknown which of the interactions between the primary members of the decapping complex, if any, are

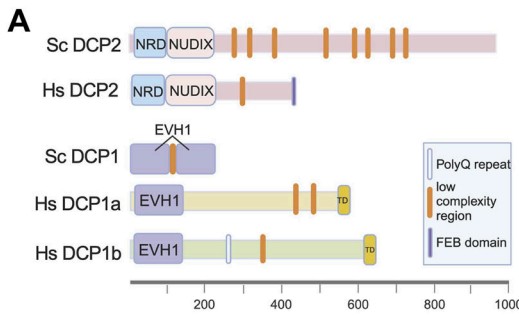

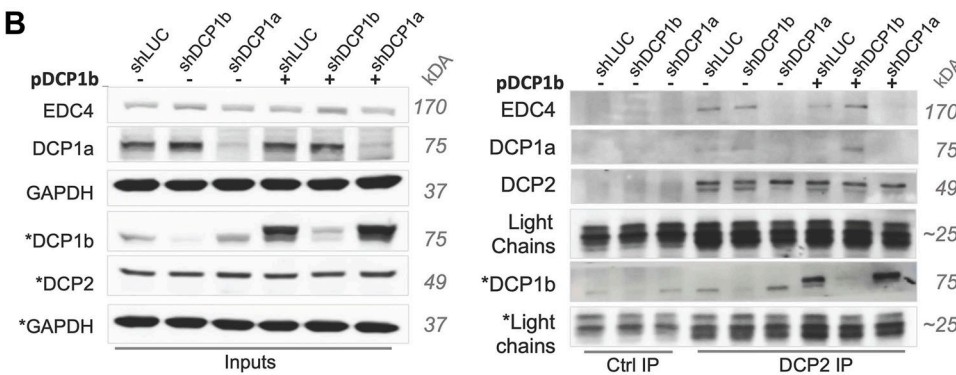

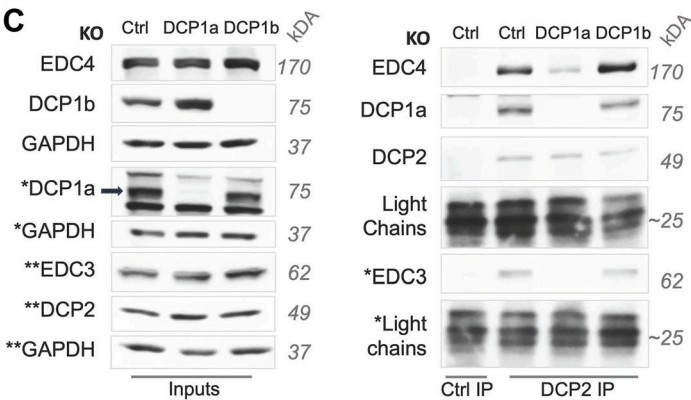

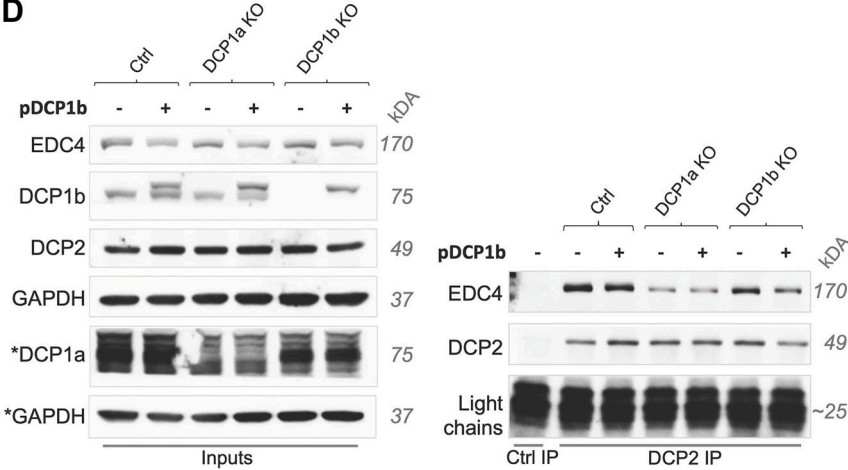

**Figure 1. DCP1a, and not DCP1b, mediates the interaction of DCP2 with EDC4 and EDC3.**

**(A)** Comparison of DCP2 and DCP1(a/b) protein domains in *Saccharomyces cerevisiae* (Sc) and *Homo sapiens* (Hs). EVH1, Ena/Vasp homology *domain* 1; NRD, N-terminus by a helical regulatory domain; TD, trimerization domain, low-complexity regions or short linear motifs. Note: The NRD and NUDIX domain in Hs DCP2 are connected by a flexible hinge (not shown) consisting of 4 amino acids. *The low-complexity regions were defined using* http://smart.embl-heidelberg.de/. *Major protein domains were defined as described previously in Jonas and Izaurralde, 2013.* Created with BioRender.com. **(B)** HCT116 wt cells were infected with lentiviruses carrying shRNAs targeting LUC, DCP1b, or DCP1a. On day two, the cells were selected with puromycin. On day 5, cells were infected with the second virus expressing DCP1b or an empty vector control. Cells were harvested 3 d after the second round of infections, and DCP2 was immunoprecipitated. Western blots (WB) showing inputs, IgG, and DCP2 immunoprecipitation in shLUC, shDCP1a, and shDCP1b cells (n = 2). * Indicates that those protein bands were from a different SDS–PAGE gel. **(C)** HCT116 cells were transfected with Cas9 protein and no gRNA to create control cells, or Cas9 protein and a mix of sgRNAs targeting either DCP1a or DCP1b. 3 d post-transfection, single-cell clones were selected and cultured for 10 d. Clones were validated and DCP2 was immunoprecipitated. WB showing inputs, IgG immunoprecipitation in control (ctrl), DCP2 IP in ctrl, DCP1a KO, and DCP1b KO cells (n = 3). * Indicates that those protein bands were from a different SDS–PAGE gel. **(D)** Ctrl, DCP1a KO (polyclonal pool), and DCP1b KO (single-cell clone) cells were infected with a lentivirus expressing DCP1b or the empty vector control. 3 d post-infection, cells were harvested and DCP2 was immunoprecipitated. WB show inputs, IgG, and DCP2 IP (n = 1). * Indicates that those protein bands were from a different SDS–PAGE gel.

Source data are available for this figure.

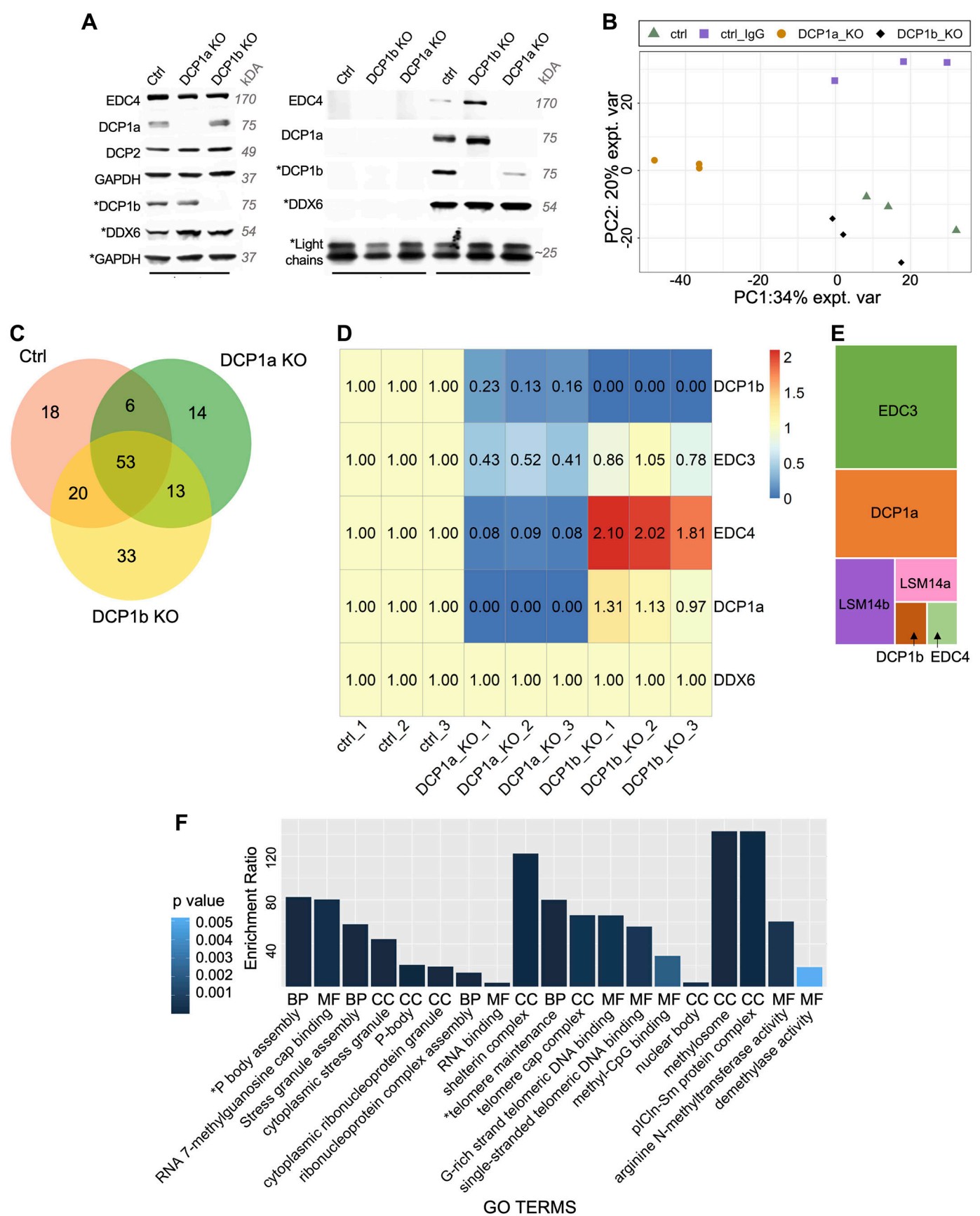

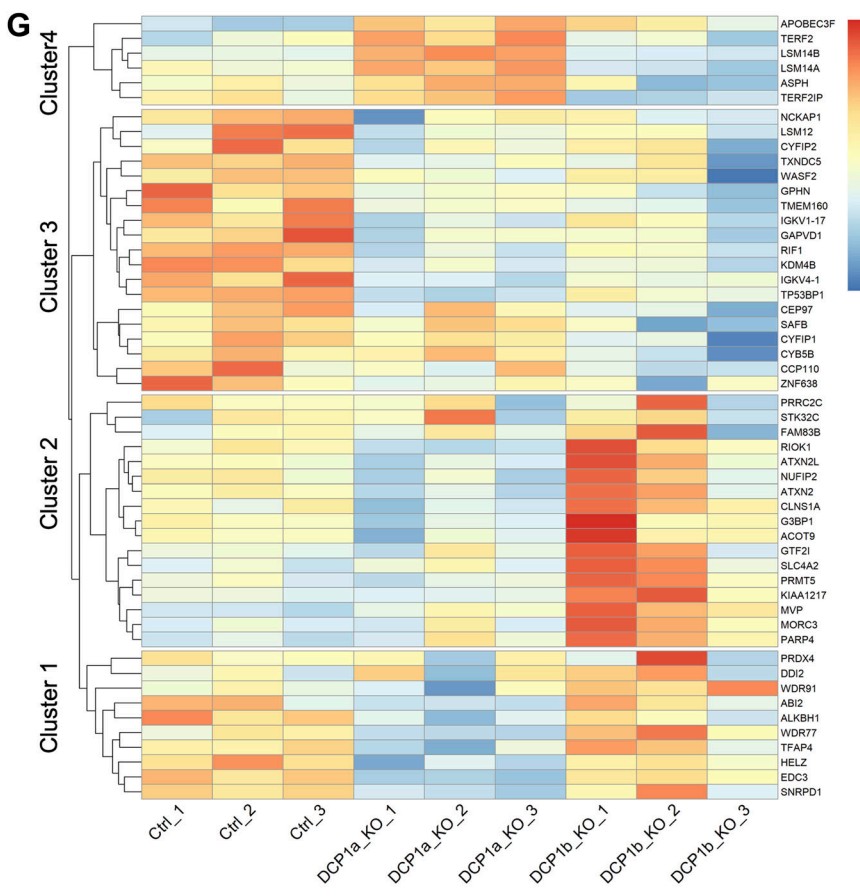

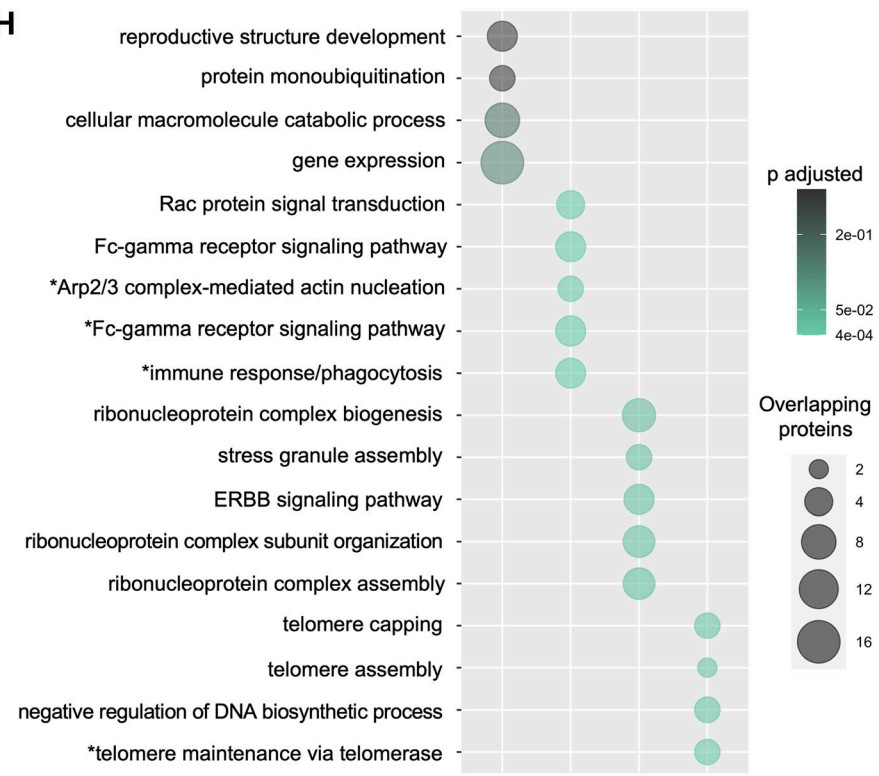

**Figure 2. DCP1a is critical for interactions between known decapping complex members.**

**(A)** DDX6 was immunoprecipitated in control cells, DCP1a KO, and DCP1b KO cells with IgG immunoprecipitation as a control. **(B)** DDX6 was immunoprecipitated in control cells, DCP1a KO, and DCP1b KO cells with IgG immunoprecipitation as a control. IPs were sent for profiling by LC/MS/MS. PCA analysis of the samples is shown. **(C)** Venn diagram represents the high-confidence proteins found in all conditions of the DDX6 IP. **(D)** Heatmap reports intensities (sum of peptide peaks as analyzed by mass spectrometry) of the core decapping complex proteins DCP1a, DCP1b, and EDC3 in the DDX6 IP for each condition. Biological replicas 1, 2, and 3 are shown. The heatmap is scaled to DDX6 intensity and ctrl samples (set to 1). The numbers in the heatmap reflect relative scaled protein intensities in these samples. Detailed explanation for normalization is available in the Materials and Methods section. Note: DCP1a and DCP1b proteins in DCP1a KO and DCP1b KO cells are not considered high-confidence by the same definition in the Methods section. The number of their unique razor peptides in those conditions is 0 as their genes are deleted. EDC4 is not considered high-confidence in DCP1a KO cells as its q-value is higher than 0.05. However, its $P$-value is lower than 0.05, and it was important to include EDC4 in this specific comparison and heatmap. **(E)** iBAQ analysis shows the relative amounts of DCP1a, DCP1b, EDC3, and EDC4 to DDX6 in the control condition. iBAQ values were used to determine the relative amounts of proteins in the DDX6 IP in the ctrl condition. Three replicas were averaged, and block size corresponds to the relative amount of DCP1a, DCP1b, EDC3, and EDC4 to DDX6 in the ctrl condition. **(F)** Ontology analysis of the core 52 proteins present in the DDX6 IP (DDX6 protein was excluded from this list). MF, Molecular Function; BP, Biological Process; CC, Cellular Compartment. Raw $P$-value is shown. Enrichment ratio incorporates the number of overlapping proteins found in the input protein set versus the GO group. **(G)** Heatmap showing intensities of the core 52 proteins (DDX6 protein was excluded from this list) that do not depend on the presence or absence of DCP1a or DCP1b in the DDX6 IP. Biological replicas 1, 2, and 3 are shown. The heatmap is created and clustered with the pheatmap function in R studio. It is scaled by row. **(G, H)** Network analysis (WebGestalt) of the clusters identified in panel **(G)**. The input proteins are used as seeds, and the top-ranking/interacting neighbors are identified and used to identify the appropriate network of proteins.

Source data are available for this figure.

supported by DCP1a versus DCP1b. To directly assess the relative roles of DCP1a and DCP1b in decapping complex assembly, DCP2 was immunoprecipitated from parental HCT116 cells and from HCT116 cells depleted of either DCP1a or DCP1b via short hairpin RNAs (shRNA). Approximately 50% of the DCP1a (Fig S3A) and 70% of DCP1b protein (Fig S3B) was depleted by shRNA treatment. Immunoblotting of the DCP2 immunoprecipitates demonstrated that depletion of DCP1a resulted in decreased interaction between DCP2 and EDC4, whereas DCP1b depletion did not affect this interaction (Fig 1B). This finding supports distinct roles for DCP1a and DCP1b in the decapping complex. However, it does not address the possibility that DCP1a and DCP1b have similar functions, but that cellular DCP1b expression levels are too low to compensate for the depletion of DCP1a, that is, a quantitative distinction rather than a qualitative one. To assess this, DCP1b was ectopically expressed at elevated levels in DCP1a-depleted cells (Fig S3). Immunoprecipitation of DCP2 from DCP1a-depleted cells showed that ectopic overexpression of DCP1b failed to rescue the interaction between DCP2 and EDC4 (Fig 1B), providing additional support for a model in which DCP1a and DCP1b have intrinsically distinct functions unrelated to their relative expression levels.

In the shRNA studies described above, DCP1a and DCP1b levels were not fully depleted (Fig S3). Whereas these studies were informative, it was also important to assess the differential roles of DCP1a and DCP1b in cells with complete loss of these gene products. For this purpose, single-cell clones where DCP1a (DCP1a KO) or DCP1b (DCP1b KO) had been targeted for deletion using CRISPR-Cas9 were isolated, and the loss of the respective protein was confirmed (Fig 1C). The cells were validated as single-cell KO clones (Supplemental Material). As in shRNA-treated cells, complete deletion of DCP1a resulted in a defective interaction between DCP2 and EDC4, whereas DCP1b deletion had no effect on this interaction (Fig 1C). Consistent with this, the decapping complex subunit EDC3 was similarly impacted by DCP1a loss, but not by DCP1b loss (Fig 1C). As in shRNA-treated cells, ectopic DCP1b overexpression did not rescue these interactions (Fig 1D). These data further support a model that includes distinct roles for DCP1a and DCP1b in the assembly and integrity of the mRNA decapping complex.

## DCP1a is critical for interactions between known decapping complex members

As targeted analysis revealed that loss of DCP1a and DCP1b have distinct effects on the interaction between DCP2 and EDC3/EDC4, an unbiased approach to defining the decapping complex interactome was taken. The RNA helicase, DDX6, interacts with DCP1a and DCP1b (39, 40, 41), allowing for the interrogation of decapping complex integrity without disturbing interactions among the core complex members (Fig 2A). To address these observations, the interactome of DDX6 in parental, DCP1a KO, and DCP1b KO was examined by proteomic analysis (Fig 2B–H).

First, empirical analysis of the DDX6 interactome revealed that DCP1b loss enhances the interaction between DCP1a and EDC4. DCP1a depletion also decreased the interaction between DDX6 and EDC4 and between DDX6 and DCP1b (Fig 2A), consistent with the

findings reported in Fig 1. In replicate experiments, immunoprecipitation and quantitative *liquid chromatography with tandem mass spectrometry* (LC/MS/MS) analysis identified 2,069 DDX6 interactors. Proteins were considered high confidence if they had (1) a minimum absolute fold change of two when compared with control, (2) a q-value < 0.05, (3) been identified by a minimum of 2 razor + unique peptides in one of the sample groups compared, and (4) been detected in at least two of the replicates in one of the sample groups compared. A principal component analysis (PCA) was used to cluster the three replicates from each condition (Fig 2B), demonstrating a high degree of concordance. This analysis revealed that the interactome of DDX6 was significantly impacted by the presence or absence of DCP1a or DCP1b (Fig 2C). In control cells, DDX6 interacts with EDC3, EDC4, DCP1a, and DCP1b, with no DCP2 being detected (Fig 2D). Deletion of DCP1a reduced the interaction of both EDC3 and EDC4 with DDX6 (Fig 2D). Conversely, the DDX6-EDC4 interaction was enhanced in DCP1b-depleted cells (Fig 2D). These findings are consistent with the empirical immunoprecipitation data shown in Fig 1. One interpretation of these findings is that DCP1a acts as a positive regulator of decapping complex assembly. The role of DCP1b in the assembly of the decapping complex is multifaceted with no discernable impact on the DDX6-EDC3 interaction, but a negative impact on the DDX6-EDC4 interaction.

Given these findings, it was of interest to determine the relative impact of DCP1a and DCP1b on the association of DDX6 with core decapping complex members. For this analysis, intensity-based absolute quantification (iBAQ) values were calculated by dividing the sum of the tryptic peptides found by the number of theoretically observable peptides. iBAQ values thereby provide an accurate estimate of the relative amount of protein within the sample (42). iBAQ values were determined for DCP1a, DCP1b, EDC4, EDC3, LSM14a, and LSM14B in a DDX6 IP from parental HCT116 cells, and DDX6 iBAQ value was set to 1. EDC4 and DCP1b are present at relatively low levels in the IP (Fig 2E). There is eightfold more DCP1a associated with DDX6 than DCP1b/EDC4 (Fig 2E). Also, striking was that the amount of EDC3 is 1.5-fold higher than that of DCP1a (Fig 2E). EDC4 is essential for decapping complex function in mammalian cells, and the finding that EDC4 and DCP1b are at limiting protein levels in the DDX6 complex may suggest a regulatory role. These findings are also consistent with DCP1a as the predominant member of the heterotrimer formed with DCP1b.

A core set of 53 proteins interacts with DDX6 regardless of DCP1a or DCP1b loss (Fig 2C). Ontology analysis showed the core 53 proteins are enriched in PB and stress granule proteins, telomere regulating proteins, and proteins with demethylase activity (Fig 2F). An intensity heatmap of these core 53 proteins was organized into four main clusters (Fig 2G). DCP1a loss decreased the interactions of DDX6 with other RNA-regulating proteins, whereas DCP1b loss did not (Fig 2G). Cluster 1 ontology analysis results are shown for proteins with a *P*-value above 0.05. In cluster 2, DCP1b loss enhanced DDX6 interactions with these 10 core proteins, whereas, in clusters 3 and 4, DCP1b loss decreased the interaction of DDX6 with nearly half of the core proteins (Fig 2G). Perhaps reflective of some functions requiring the cooperative activity of DCP1a and DCP1b, a topological network analysis of the clusters shows that cluster 3,

where loss of DCP1a and DCP1b decreases DDX6 interactions, has a significant number of proteins related to the stress granule/ribonucleoprotein complex (Fig 2H). Network analysis of cluster 4, where loss of DCP1a and loss of DCP1b have opposing effects on DDX6 interactions, is enriched in telomere capping and maintenance proteins (Fig 2H). The presence of both repressed and enhanced DDX6 interactions underscores the distinct roles of DCP1a and DCP1b in the decapping complex.

Distinct groups of proteins were identified whose interaction with DDX6 depends on the presence of DCP1a/b or both. These are the proteins whose interaction with DDX6 is lost upon DCP1a/b depletion, specifically, 6 DCP1a-dependent proteins, 20 DCP1b-dependent proteins, and 18 DCP1a- and DCP1b-dependent proteins. Ontology analysis revealed that DCP1a-dependent proteins regulate synaptic vesicle exocytosis, deadenylation-dependent and -independent decapping of mRNA, and mRNA cap-binding (Fig S4). The DCP1b-dependent proteins were enriched in translation initiation regulators, mRNA splicing, and nonsense-mediated decay (Fig S4).

### DCP1a and DCP1b have primarily independent interactomes

To further elucidate the multifaceted roles of DCP1a and DCP1b, these proteins were directly immunoprecipitated from parental HCT116 cells, and from DCP1a KO and DCP1b KO cells (Fig 3A and B). DCP1a KO cells are a polyclonal pool where 25%, compared with control cells, of DCP1a is still present (Fig 3), and DCP1b KO cells are derived from a single-cell clone where DCP1b is fully deleted (Supplemental Material). Proteins were considered high confidence if they were: (1) identified by a minimum of 2 razor + unique peptides in the experimental condition, (2) identified in both experimental replicas and (3) had a minimum fold change of 2 (if not identified in the IgG control) or 20 (if the protein is also identified in the IgG control). Based on epitope mapping, the antibodies used for DCP1a and DCP1b are not expected to interfere with their (hetero)trimerization nor their binding to DCP2 (30, 36). LC-MS/MS analysis revealed that DCP1a interacts with 88 high-confidence proteins. Upon loss of DCP1b, DCP1a retains its interaction with 73 of these 88 proteins and acquires the ability to interact with 21 new partners (Fig 3C). Using the same criteria, DCP1b interacts with 385 proteins. Upon loss of DCP1a, DCP1b still interacts with 271 of these 385 proteins, when also acquiring 203 new partners (Fig 3C). In parental cells, only six proteins were present in both the DCP1a and DCP1b IPs (DCP1b, DCP1a, EDC3, SLIRP, TBL1XR1, TERF2IP). Of note, DCP1a and DCP1b were present in both IPs, along with the core decapping complex member EDC3. As shown previously, even partial loss of DCP1a impacts the interaction of EDC3 with DCP1b, whereas complete loss of DCP1b does not perturb the EDC3-DCP1a interaction. The largely distinct nature of the DCP1a and DCP1b interactomes further supports a model in which DCP1a and DCP1b have distinct cellular roles.

To gain a more thorough understanding of the roles of DCP1a and DCP1b, their relative protein levels in cells were compared using iBAQ values (Fig 3C). In the DCP1a IP, DCP1a and DCP1b were present in ratios of 1:0.3 and 1:0.31 in the two replicates, whereas in the DCP1b IP, the DCP1b:DCP1a ratio in two replicates was 1:0.50 and 1:0.58. The relative amount of DCP1b to DCP1a in the DCP1b IP is

consistent with heterotrimers containing two DCP1b subunits and a single DCP1a subunit. This contrasts with what iBAQ analysis indicated for the DDX6 IP. These divergent findings may be explained by DCP1a and DCP1b being present in the decapping complex at different ratios in different contexts. To assess this question, gel filtration was used to examine the molecular weight distribution of decapping complex subunits. Whole-cell lysates from control, DCP1a KO, and DCP1b KO cells were fractionated over a Superose 6 size exclusion column, and eluted fractions were examined by Western blotting (Fig S5). This analysis revealed that decapping complex components reside in at least two distinct complexes. Specifically, they appear in fractions 17–22, defined as decapping complex v1, which is near the upper level of column resolution (~5 MD); and fractions 22–31, defined as decapping complex v2, in the linear range of separation between 700 kD–2 MD (Fig S5A–D and E). Whereas the molecular weight cannot be estimated accurately from these data, it does indicate that the decapping complex exists in at least two different assemblies of distinct size and composition. In control cells, DCP1b is present at lower levels in the decapping complex v1 relative to complex v2, whereas DCP1a protein levels are similar in both versions of the decapping complex (Fig S5E). The presence of DCP1a and DCP1b in distinct size exclusion patterns further supports the model that these paralogs have partially distinct functions.

Overall, ontology analysis revealed that DCP1a interacts with proteins that have histone and protein deacetylase activity, bind mRNA, tubulin, and chromatin, and have transcription corepressor activity (Fig 3E and F). Cellular component analysis suggested that DCP1a can be found in P bodies but also in histone deacetylase complexes, spindles, and chromatin (Fig 3E). In contrast, the DCP1b interactome is enriched in proteins that regulate translation, including proteins that have aminoacyl-tRNA ligase activity (Fig 3F). However, DCP1b also interacts with proteins that are part of the proteasome complex and bind cadherin, RNA, and purine ribonucleotides (Fig 3F). DCP1b further interacts with proteins involved in neutrophil activation, translational initiation, and focal adhesion/adherens junctions (Fig 3F). These data suggest that whereas DCP1a and DCP1b have essential roles within the decapping complex, they likely perform additional cellular functions.

### DCP1a and DCP1b are part of the transcript buffering feedback system

EDC3 and EDC4 regulate the efficiency and kinetics of decapping (43, 44), and DCP2 can impact the degradation rates of specific groups of mRNAs (45, 46, 47). As DCP1a and DCP1b impact decapping complex interactions involving EDC3, EDC4, and other subunits, experiments were designed to examine their impact on the half-lives of mRNAs. To examine the effects that DCP1a and DCP1b have on global RNA half-life, *thiol(SH)-linked alkylation for the metabolic sequencing of RNA* (SLAM-seq) was performed (48). Control, DCP1a KO, and DCP1b KO cells were cultured with 4SU for 24 h and then 4SU RNA was diluted during a 24 h chase, while cells were collected at 0, 4, 8, and 24 h after labeling was stopped (Fig 4A). In SLAM-seq, transcriptome-wide RNA sequencing detects 4-thiouridine (4SU) incorporation into RNA at single-nucleotide resolution, based on

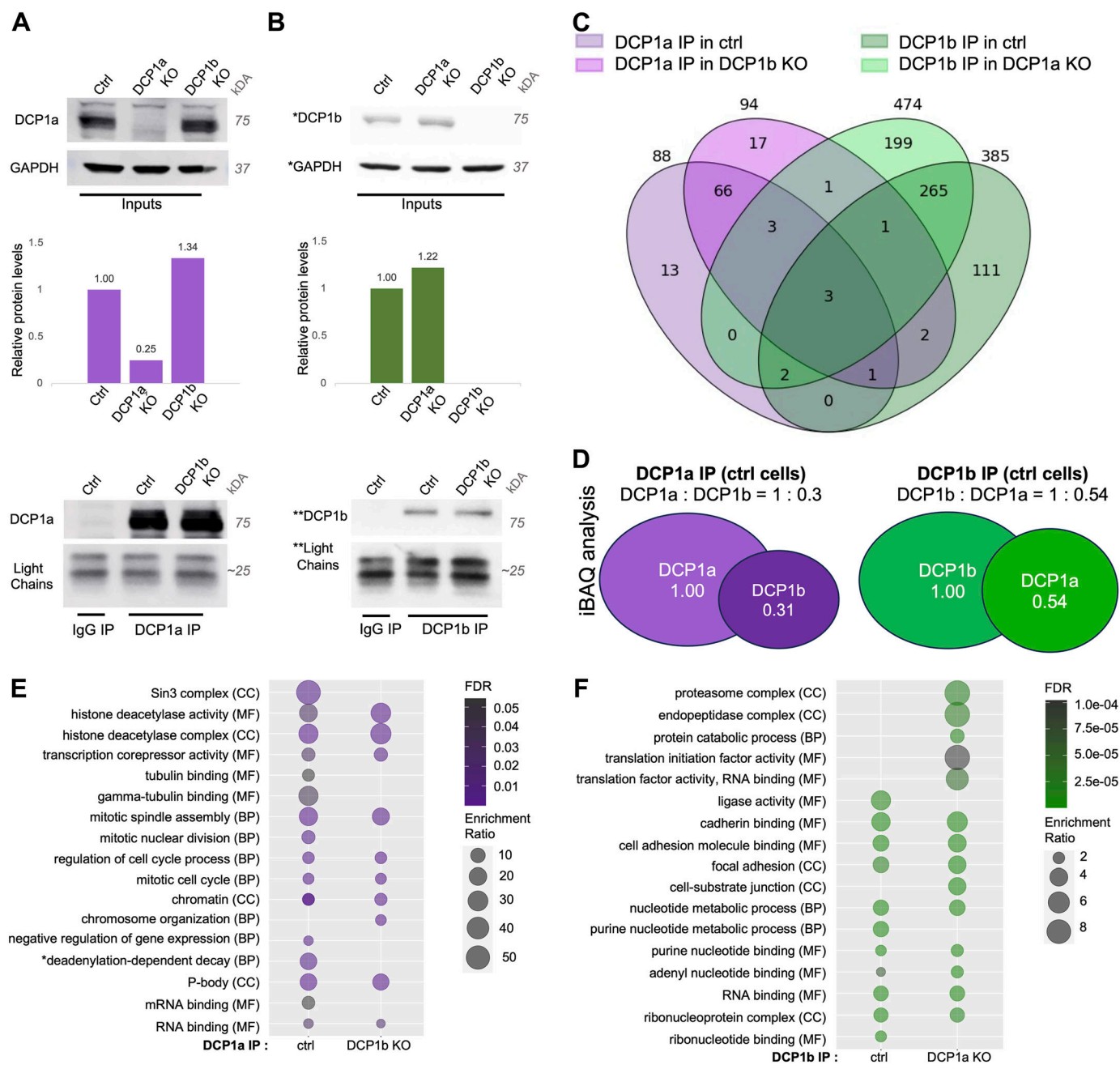

**Figure 3. DCP1a and DCP1b have primarily non-overlapping interactomes.**
**(A, B)** DCP1a was immunoprecipitated from HCT116 cell lysates (ctrl) and DCP1b KO cells. **(B)** DCP1b was immunoprecipitated from HCT116 cell lysates (ctrl) and DCP1a KO cells. *, ** indicate different SDS–PAGE gels. The IPs on these Western blots are from replica 1, which was used for the creation of the LC/MS/MS datasets in this figure. **(C)** Venn diagram showing high-confidence proteins identified in DCP1a and DCP1b IPs. **(D)** iBAQ analysis showing relative levels of DCP1a to DCP1b in their respective IPs in ctrl cells. **(E, F)** Relevant GO terms from ontology analysis of (**E**) DCP1a IPs in ctrl and DCP1b KO condition and (**F**) DCP1b IP in ctrl and DCP1a KO condition. Enrichment ratio incorporates the number of overlapping proteins found in the input protein set versus the GO group. MF, Molecular Function; BP, Biological Process; CC, Cellular Compartment.*GO terms have been edited for brevity. Full GO analysis can be found in the supplemental data section.
Source data are available for this figure.

T > C conversions ([48]) (Fig 4A). Conditions were optimized by evaluating the impact of 4SU concentrations on cell viability (Fig S6A). Cellular uptake of 4SU was confirmed by dot blot analysis of 4SU incorporation into RNA (Fig S6B). mRNA libraries were sequenced at 75 bp single read resolution (NextSeq 500; Illumina) and analyzed using SLAM-DUNK ([48]) and GRAND-SLAM algorithms ([49]). SLAM-DUNK was used to assess the T>C conversion rates, normalized across UTRs (Supplemental Material). Genes are retained if they have at least 100 reads in all the samples. Half-lives and synthesis rates are estimated using GrandR ([50]).

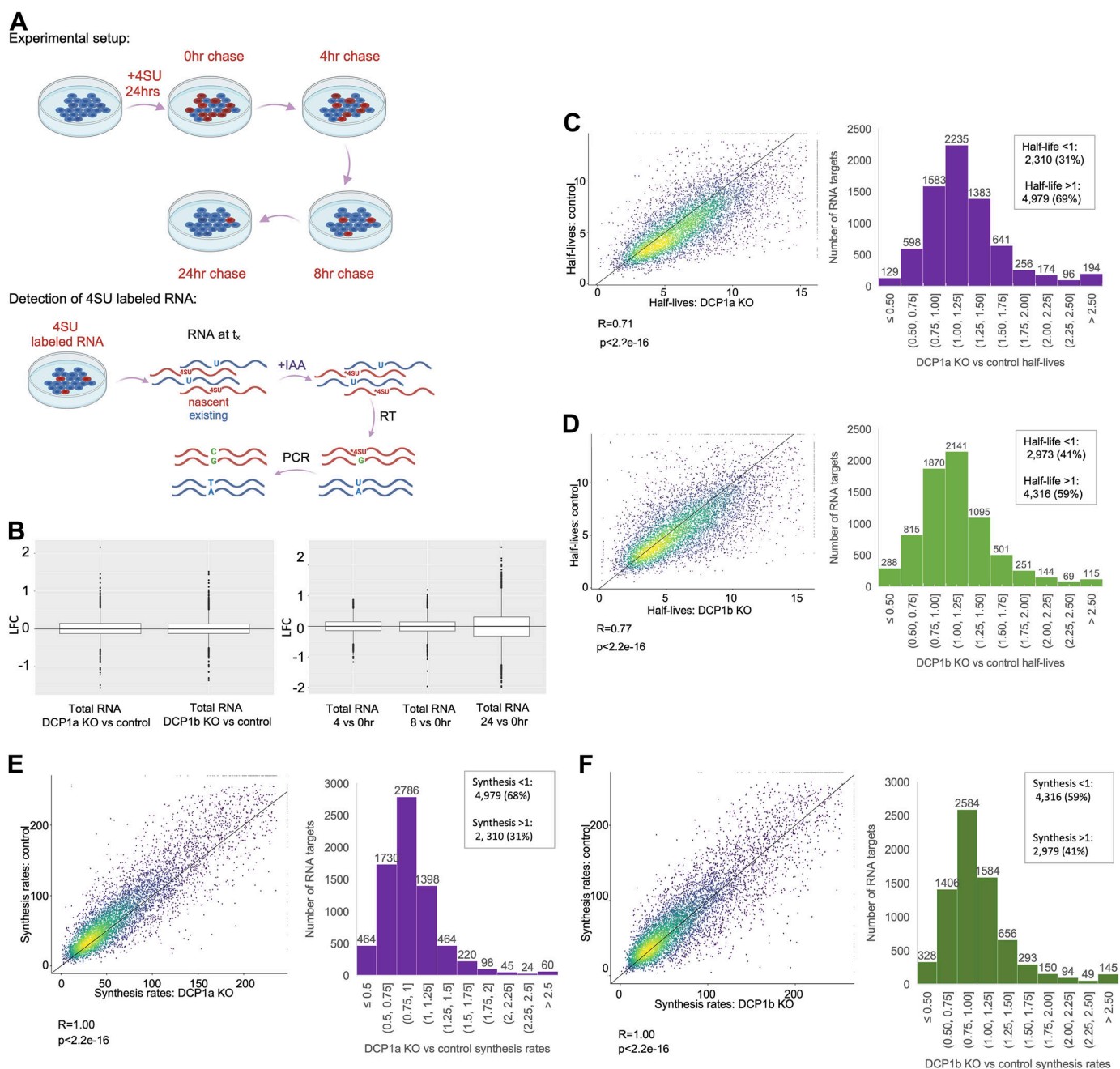

**Figure 4. The impact of DCP1a and DCP1b on mRNA half-life may be attenuated by effects on transcript buffering.**
**(A)** Experimental pipeline for determining mRNA $t_{1/2}$ using thiol(SH)-linked alkylation for the metabolic sequencing of RNA (SLAM-seq). This method allows for direct quantification of 4-thiouridine (4SU) within the 4SU labeled mRNA. Control, DCP1a KO, and DCP1b KO cells were cultured with 100 $\mu$M 4SU for 24 h, whereas fresh 4SU media was provided every 3 h and cells were kept in a light-free environment. 4SU content was then diluted during the chase. Cells were collected at 0, 4, 8, and 24 h after the end of labeling. Cells were harvested and RNA was isolated and alkylated in a light-free environment. Alkylated RNA was purified and mRNA libraries were prepared and subjected to RNA-seq. When 4SU, as a uracil analog, is incorporated and alkylated, reverse transcriptase misincorporates guanosine. The 4SU content of mRNA was quantified by measuring the T > C conversions in the final 3' end mRNA sequencing. Samples were sequenced at a depth of 75 bp single read sequencing. **(B)** Total RNA

The GrandR pipeline has built-in functions (log fold change and pairwise DESeq) to allow assessment of the total RNA changes over the course of the experiment (51). This analysis revealed no significant impact on total RNA levels between conditions (DCP1a/DCP1b KO versus control) (Fig 4B). Stability and synthesis rates were quantified for 7,289 transcripts in all three conditions, where the average half-lives were 6.54, 7.84, and 7.04 h for control, DCP1a KO, and DCP1b KO conditions, respectively (Fig 4C and D). On average, both DCP1a and DCP1b loss lead to increased mRNA half-life relative to control cells. Specifically, 4,979 (69%) and 4,316 (59%) of the quantified RNAs have a longer half-life (the ratio of DCP1a KO or DCP1b KO to control half-life is greater than 1) upon DCP1a and

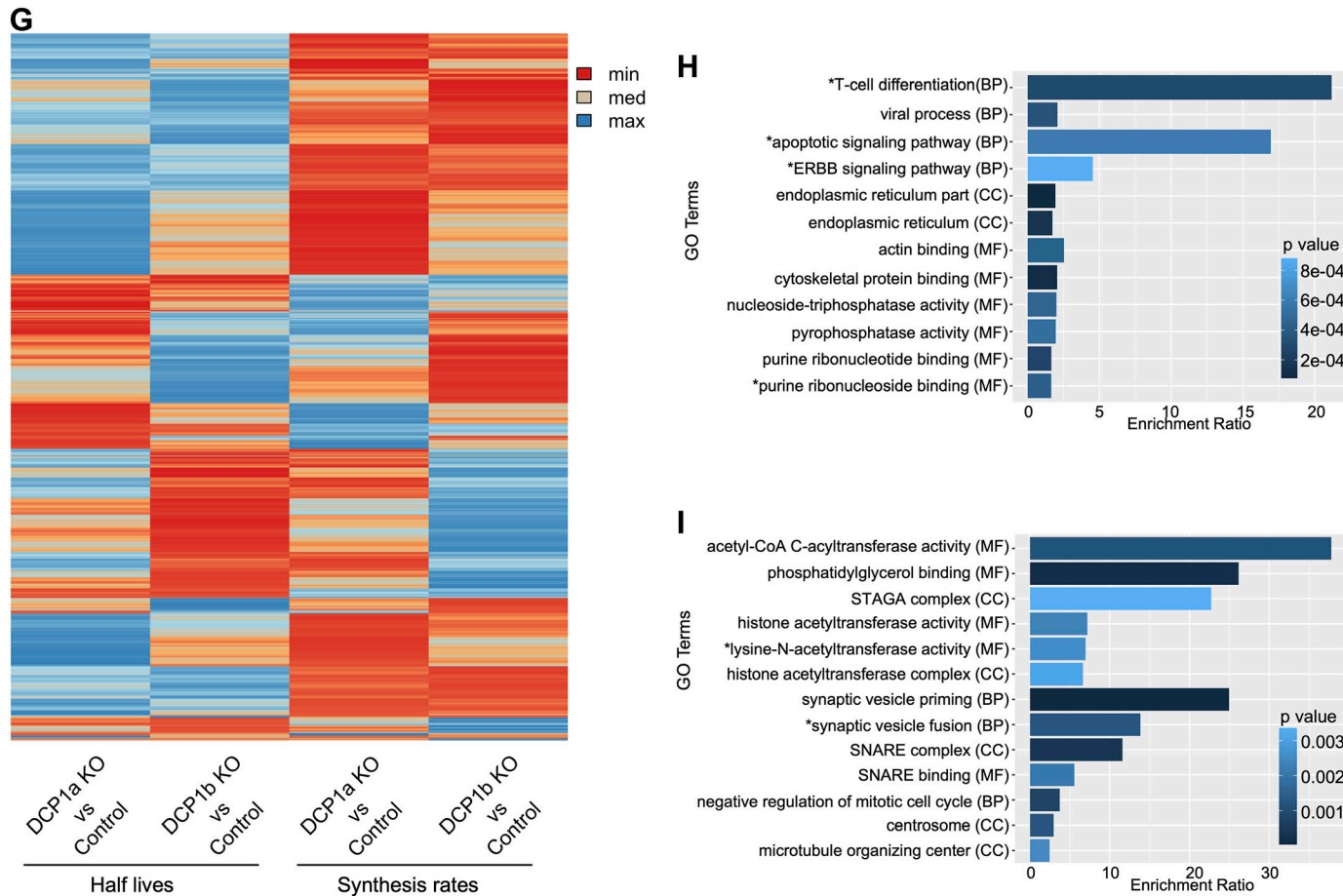

**Figure 4. (Continued)** change between the conditions and timepoints using LFC estimations as described. **(C, D)** Scatterplots with Pearson correlation coefficients, associated *P*-values, and histograms of the half-lives estimated in **(C)** DCP1a KO versus control and **(D)** DCP1b KO versus control. **(E, F)** Scatterplots with Pearson correlation coefficients, associated *P*-values, and histograms of the synthesis rates estimated in **(E)** DCP1a KO versus control, and **(F)** DCP1b KO versus control. **(G)** Heatmap showing half-lives and synthesis rates in DCP1a KO versus control and DCP1b KO versus control. **(H, I)** Over-representation ontology analysis showing relevant GO terms for identified **(H)** DCP1a- and **(I)** DCP1b-dependent mRNAs. Only relevant GO terms are shown; full analysis is supplied as supplementary material. Enrichment ratio is normalized and incorporates the number of overlapping targets and the number of targets in the pathway/GO term. *GO terms have been edited for brevity. MF, Molecular Function; BP, Biological Process; CC, Cellular Compartment.

DCP1b loss, respectively (Fig 4C and D). Although on average, half-lives increased, synthesis rates decreased upon DCP1a and DCP1b loss (Fig 4E and F). DCP1a plays a significant role in decapping complex assembly, but both DCP1a and DCP1b impact decay and synthesis of mRNAs. The decapping complex has been implicated in transcript buffering, a process in which mRNA levels are kept constant through reciprocal changes in mRNA synthesis and degradation (23, 24, 52). The data presented here demonstrate for the first time that DCP1a and DCP1b have an impact on mRNA synthesis rates. Furthermore, the impact that they have on synthesis rates is reciprocal to their impact on half-lives, underlining the role that DCP1a and DCP1b play in transcript buffering (Fig 4G). Based on the assumption that loss of DCP1a/b should decrease mRNA decapping and thereby increase mRNA stability, direct mRNA targets are predicted to have significantly longer half-lives in the absence of DCP1a or DCP1b. Therefore, stabilized RNAs were defined as DCP1a or DCP1b dependent. DCP1a and DCP1b up-regulate half-lives of the same groups of

mRNAs and that of the distinct groups of mRNAs. DCP1a significantly impacts half-lives of 374 targets (ratio of DCP1a KO to control half-life > 1.5, and DCP1b KO to control half-life < 1). DCP1b significantly impacts half-lives of 195 RNAs (ratio of DCP1b KO to control half-life > 1.5, and DCP1a KO to control half-life < 1). There are 416 mRNAs that are significantly impacted by the loss of either DCP1a or DCP1b (DCP1a/DCP1b KO versus control half-life greater than 1.5). Of note, DCP1a-dependent RNAs are linked to T-cell differentiation, purine ribonucleotide binding, and regulation of nucleoside-triphosphatase activity (Fig 4H), whereas DCP1b-dependent RNAs are linked to histone acetylation and synaptic vesicle fusion (Fig 4I).

A previous study identified a subset of RNAs as stabilized upon DCP2 loss (45). Specifically, they identified 1,804 DCP2-dependent mRNAs. A subset of these established DCP2 targets are also stabilized in the absence of DCP1a or DCP1b. Specifically, 717 and 626 of the 1,804 DCP2 target mRNAs are mildly stabilized in the absence of DCP1a or DCP1b, respectively (ratio of control to

KO condition half-life > 1). Furthermore, a subset of DCP1a- and DCP1b-dependent mRNAs, are also DCP2 targets. Specifically, 69/374 and 36/195 of the DCP1a- and DCP1b-dependent mRNAs are also DCP2 targets. Ontology analysis revealed that DCP1a-modulated DCP2 targets encode proteins that are part of the ESCRT I complex and endosome membrane, but interestingly also impact transcription (Fig S7). DCP1b-modulated DCP2 targets are part of the CCR4-NOT core complex, suggesting some cross talk between deadenylation and decapping. DCP1b modulated DCP2 targets that encode proteins that are also a part of the histone acetyl transferase complex and impact transcription (Fig S7). It is notable that although distinct, both DCP1a- and DCP1b-modulated DCP2 targets play a role in transcription regulation pathways.

To validate these results, progressive labeling of RNAs (0 and 4 h) with 4SU was performed. The impact of different 4SU concentrations on cell viability was assessed (Fig S6C) and cellular uptake of 4SU was confirmed via dot blot analysis of 4SU incorporation into RNA (Fig S6D). mRNA libraries were sequenced at 75 bp single read resolution (NextSeq 500; Illumina) and analyzed using SLAM-DUNK (48), GRAND-SLAM (49), and GrandR algorithms (50). The MultiQC module of SLAM-DUNK was used to assess the T>C conversion rates, normalized across UTRs (Supplemental Material). Half-lives and synthesis rates were estimated using GrandR (50). Genes were considered for further analysis if they had at least 200 read counts in all the samples and were normalized by DESeq2 estimated size factors.

Total RNA did not significantly change between conditions or upon treatment with 4SU (Fig S8A). As long half-lives cannot be accurately estimated with a 4 h labeling time, an arbitrary half-life cutoff was made at 20 h in control cells (Fig S8B and C). The dataset was further filtered to include only the genes with an estimated half-life under 20 h in the control condition (Fig S8D–G). After filtering, the impact of DCP1a and DCP1b on mRNA half-life regulation and synthesis was examined by analyzing 2,690 RNAs (Fig S8D–G). As in the pulse-chase experiment above, results show that loss of either DCP1a or DCP1b generally prolongs mRNA half-life. Specifically, 1,682 (62%) and 2,150 (80%) of the mRNAs had increased half-lives upon DCP1a or DCP1b loss, respectively. Those targets also exhibited decreased synthesis rates upon DCP1a or DCP1b depletion. These results are consistent with DCP1a and DCP1b impacting both decapping and transcription to maintain steady-state levels of mRNA.

Having determined the impact of DCP1a and DCP1b on mRNA half-lives via two different labeling strategies (pulse-chase and progressive), provided the opportunity to increase confidence in the results by comparing the findings across the two methods. Of the 2,690 mRNAs identified in the progressive labeling experiment, 2,470 were also identified in the pulse-chase experiment. Of these 2,470 mRNAs, 1,729 were stabilized by the loss of DCP1a in the pulse-chase experiment. Most of these stabilized mRNAs (1,079) were also stabilized in the progressive labeling experiment. Similar concordance between the pulse-chase and progressive labeling experiments was found for DCP1b KO cells. Here, the pulse-chase experiment demonstrated that 1,499 mRNAs were stabilized by DCP1b loss and 1,199 of these were also stabilized in the progressive labeling experiment.

## DCP1a and DCP1b impact the stability of proteins involved in maintenance of chromosomal stability

Finally, the impact of DCP1a or DCP1b loss on the overall proteome was examined. For this purpose, protein lysates were generated from parental, DCP1a KO, and DCP1b KO cells and subjected to quantitative profiling by LC/MS/MS. A total of 5,543 proteins were identified. Proteins identified in DCP1a KO and DCP1b KO cells were compared with those identified in control cells. PCA analysis demonstrated a general clustering of the samples (Fig 5A) and only high-confidence proteins were subjected to further analysis (869 proteins were identified as high confidence in DCP1b KO, 741 proteins in DCP1a KO, and 238 high-confidence proteins in both DCP1a KO and DCP1b KO cells). Most of the proteins in DCP1a KO and DCP1b KO cells had a log2 ratio of less than 1 or higher than −1, illustrating that on the protein level, with DCP1a and DCP1b loss, there are moderate changes (Fig 5B and C). Specifically, 44 proteins were strongly impacted by DCP1a loss (Fig 5B) and 38 proteins by DCP1b KO (log2 ratio greater than 1 or less than −1) (Fig 5C). DCP1a loss significantly impacted proteins that are linked to DNA replication and associated with the positive regulation of telomerase activity (Fig 5D). Conversely, DCP1b loss significantly impacted levels of proteins linked to the KEOPS complex, which has a role in telomere uncapping and maintenance of chromosomal stability (53) (Fig 5E).

# Discussion

Regulation of mRNA decapping plays a critical role in gene expression. A single DCP1 cofactor modulates the activity of the decapping complex catalytic subunit DCP2 in lower eukaryotes such as yeast. In higher eukaryotes, including humans, two paralogous cofactor proteins, DCP1a and DCP1b, work with DCP2. The studies reported here are the first to define the key functional distinctions between the human DCP1 paralogs. Despite both containing the conserved EVH1 protein domain, DCP1a and DCP1b differ in terms of protein interactomes and in their role in regulating the integrity of the mRNA decapping complex itself (Figs 1, 2, and 3). In the cell platform used here, DCP1a is likely over-represented in the heterotrimeric DCP1a/b module of the decapping complex (Figs 2E and 3C). DCP1a and DCP1b interactomes are largely non-overlapping, emphasizing their unique cellular roles (Fig 3). Functionally, the transcripts impacted by DCP1a loss are largely distinct from those impacted by DCP1b loss (Fig 4D and F). Evidence presented here also shows that DCP1a and DCP1b loss impacts mRNA synthesis rates, potentially implicating the decapping complex in transcript buffering, as has been suggested in yeast and human cells (54, 55, 56). Collectively, these findings demonstrate that DCP1a and DCP1b function as distinct, non-redundant cofactors of the decapping enzyme DCP2.

As mentioned above, DCP1a and DCP1b have unique roles in decapping complex integrity. For example, DCP1a impacts the interaction of DCP2 with EDC4 and EDC3 (Figs 1B–D and 2B), in a manner that cannot be compensated for by ectopic DCP1b expression (Fig 1B and D). In contrast, DCP1b plays an inhibitory role in the recruitment of EDC4 (Fig 2B). This pattern is observed among decapping complex interactions more broadly, where DCP1a depletion reduces the intensity of the DDX6 interactions, whereas the

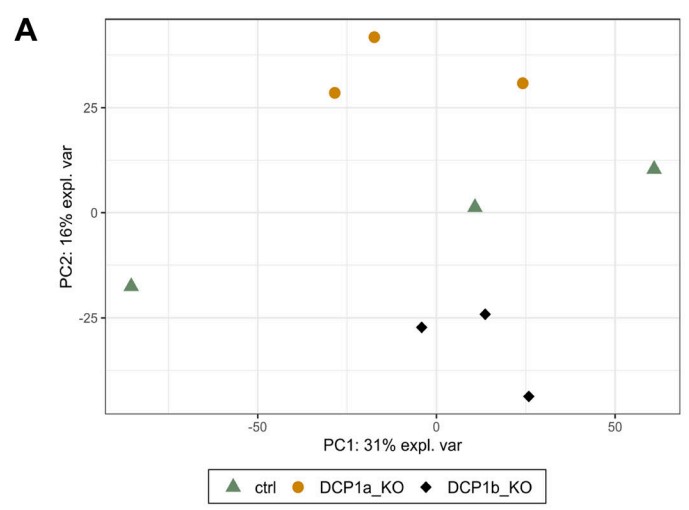

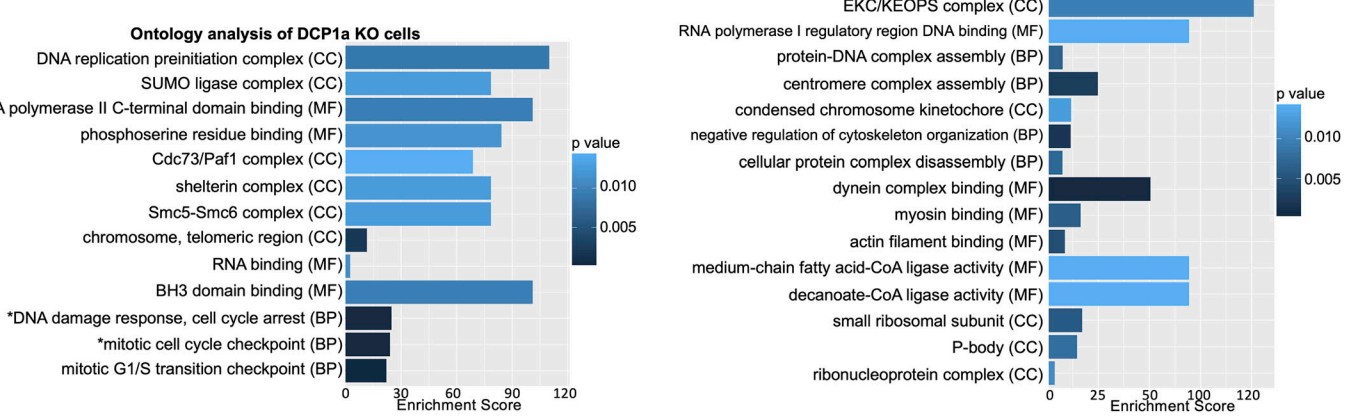

impact of DCP1b depletion is less dramatic (Fig 2G). These data suggest that DCP1a and DCP1b have reciprocal roles in decapping complex assembly/integrity, with DCP1a functioning as a positive regulator and DCP1b as a negative regulator. One possible explanation for these reciprocal roles is the differing number of low-complexity regions in DCP1a and DCP1b. Low-complexity regions mediate the strength of the decapping complex binding. DCP1a has two such regions, whereas DCP1b has only one. We know that DCP1a and DCP1b can homo- or hetero-trimerize. The assumption is that when DCP1b is depleted, the DCP1a trimer is part of the decapping complex, and vice versa. It is possible that DCP1a, because of having one extra low-complexity region, has stronger avidity effects and allows for stronger decapping complex formation. On the other hand, DCP1b alone in the complex may not be able to bring in the other decapping complex interactors and stabilize the decapping complex to the same degree. Therefore, with DCP1b loss, decapping complex interactions would appear enhanced and with DCP1a loss, decapping complex interactions would appear impaired. Because DCP1a and DCP1b can form either homo- or heterotrimers within the decapping complex, modulating the stoichiometry of DCP1a-DCP1b content could provide a cellular mechanism for regulating complex assembly and decapping activity. Furthermore, DCP1a and DCP1b appear to be present at different ratios in different tissues (Fig S1) and the decapping complex itself appears to exist in at least two macromolecular configurations (Fig S5).

As shown, DCP1a and DCP1b impact the decay rates of distinct groups of mRNAs. Specifically, 374 DCP1a-dependent mRNAs, 195 DCP1b-dependent mRNAs, and 416 DCP1a- and DCP1b-dependent mRNAs were identified (Fig 4). Over-represented among the transcripts controlled by DCP1a are transcripts that encode proteins involved in T-cell differentiation, viral processes, and purine ribonucleotide binding. DCP1b-dependent mRNAs include those encoding proteins involved in histone acetylation and synaptic vesicle fusion. As these studies were conducted in the colorectal carcinoma line HCT116, the physiological implications of this impact on T-cell–associated transcripts are unclear. Whereas this study is the first to define DCP1a- and DCP1b-dependent mRNAs, the ontology analysis is consistent with previous reports implicating DCP1a in the immune response and the decapping complex in transcription regulation (56, 57). Whereas the list of DCP1a- and DCP1b-dependent mRNA targets identified here is likely incomplete, it expands the available knowledge and provides an opportunity for a deeper analysis of distinctions between DCP1a- and DCP1b-dependent mRNAs.

As DCP1a or DCP1b loss stabilized 68% and 58% of the interrogated mRNAs, respectively, it also increased the transcription of those targets. mRNA decapping complex members have previously been implicated in establishing a transcript feedback loop that "buffers" steady-state mRNA levels (54, 56, 58), and DCP1a and DCP1b localize to both the nucleus and cytoplasm (Fig S9). From these data, a potential mechanism responsible for the indirect role of DCP1a/b in the transcript buffering system can be inferred. For example, a significant group of DCP1b-dependent mRNAs encode proteins linked to histone acetyltransferase complexes (Fig 4I). Furthermore, both DCP1a- and DCP1b-impacted RNAs, from a subset of the DCP2 target group, appear to encode proteins involved in transcriptional regulation (Fig S7). Specifically, DCP1a-dependent DCP2 mRNAs encode for proteins involved in pathways that regulate histone acetylation, whereas DCP1b-dependent DCP2 mRNAs encode proteins that are linked to transcriptional repressor complexes. On the other hand, DCP1a interacts with proteins involved in histone deacetylation, further connecting the decapping complex to specific biochemical activities that regulate eukaryotic transcription (Fig 3D). Additional analysis is required to delineating histone acetylation patterns in response to stimuli and manipulation of DCP1a and DCP1b. On the other hand, DCP1b interacts closely with proteins involved in protein degradation and translation machinery (Fig 3E), connecting the decapping complex with yet another mechanism involved in gene regulation. All these discoveries support the developing model that mRNA decapping is an essential element in the transcript buffering pathway.

It is important to note that datasets from the 4SU labeling experiments may be biased to preferentially include highly transcribed mRNAs. The SLAM-seq methodology relies on the conservative use of 4SU, which dramatically impacts cell stress signaling when used at higher concentrations. The low concentrations of 4SU used here allowed the analysis of DCP1a- or DCP1b-regulated transcripts, presumably because those are transcribed at a rate high enough to incorporate significant levels of 4SU during the labeling interval.

The data presented here indicate that the mRNA decapping complex exists in multiple configurations in human cells (Figs 3C and S5). This fluid composition of the mRNA decapping complex could provide an opportunity to differentially control distinct sets of mRNA targets. It also provides a potential mechanism for controlling the distinct functions of the decapping complex, for example, mRNA stability, translation efficiency, and/or transcript buffering. From the initial studies reported here, DCP1a is uniquely essential for efficient decapping complex assembly, interacting with RNA cap-binding proteins, and being responsible for the turnover of mRNAs encoding proteins directly involved in both adaptive immunity and transcription. In contrast, DCP1b plays a unique

**Figure 5. The impact of DCP1a and DCP1b on the cellular proteome.**
**(A)** Lysates were collected from control, DCP1a KO, and DCP1b KO cells and subjected to SDS/PAGE. Proteins were stained, and bands were excised and analyzed by LC/MS/MS. PCA analysis is shown. **(B, C)** Volcano plot shows $\log_2$(Fold change) versus $\log_{10}$($P$-value) in (**B**) DCP1a versus ctrl condition protein levels and (**C**) DCP1b versus ctrl condition of the 5,542 proteins identified. Vertical line indicates $\log_2$ fold change of 1 or −1. The horizontal line represents a $P$-value of 0.05. High-confidence proteins are shown in purple. **(D)** Ontology analysis of only significantly impacted proteins (which did not appear in DCP1b KO results) in DCP1a KO versus ctrl. GO terms with similar cellular roles are grouped. Only relevant GO terms with a $P$-value < 0.05 are shown. MF, Molecular Function; BP, Biological Process; CC, Cellular Compartment. **(E)** Ontology analysis of only significantly impacted proteins (which did not appear in DCP1a KO results) in DCP1b KO versus ctrl. Comparison of half-life of DCP1a-dependent mRNAs and $\log_2$ fold change in protein level and DCP1b-dependent mRNAs and $\log_2$ fold change in protein level.
Source data are available for this figure.

role in fostering the interaction of the decapping complex with the proteins involved in protein degradation, and the translational machinery, when simultaneously playing a role in the turnover of mRNAs encoding proteins involved in transcription.

Ultimately, it will be important to delineate how the relative stoichiometry of DCP1a and DCP1b in the decapping complex is controlled when cells are exposed to different stimuli, and how altering that stoichiometry impacts decapping activity and specificity. As the biochemical events responsible for transcript buffering are elucidated, it will also be critical to understand how DCP1a and DCP1b modulate those events.

# Materials and Methods

### Cell culture, cell treatments, lentivirus production, cell infection, transfection, and CRISPR

Human cells (HCT116 and HEK293T) were obtained from American Type Culture Collection (ATCC) and cultured in DMEM (Corning) supplemented with 10% FBS (Gemini Bio-Products) at 37°C in 5% $CO_2$. Cells were evaluated at 4-wk intervals for mycoplasma (Sigma-Aldrich). Tetracycline (Tet)-inducible p53-expressing stable cell lines (H1299To-p53) were created as described previously (59). p53 expression was induced by treating H1299To-p53 cells with 1 $\mu$g/ml of Tet (Millipore Sigma). All stock plates were treated with plasmocin (InvivoGen) as per the manufacturer's instructions.

Packaging plasmids (psPAX2 + pMD2.G) were co-transfected into HEK293T cells, along with shRNA or cDNA expression vectors, to deplete or overexpress DCP1b or DCP1a. Viral supernatants were collected, filtered, and added to the appropriate cells in the presence of 8 $\mu$g/ml polybrene (Sigma-Aldrich). Puromycin (Sigma-Aldrich) was used for the selection of infected cells. For knockdown, HCT116 cells were infected with lentiviral carrying shRNA specific to DCP1a (OriGene) or DCP1b (OriGene) and for overexpression with lentiviral expression plasmid for DCP1b (OriGene). As a control, luciferase shRNA obtained from TRC collection was used (Sigma-Aldrich).

To delete the DCP1b or DCP1a genes, HCT116 cells were transfected with the appropriate mixture of three different sgRNAs (Synthego Inc.) and Cas9 protein (Synthego Inc.), as per the manufacturer's instructions. Briefly, Cas9 was diluted to 3 $\mu$M with Opti-Mem (Thermo Fisher Scientific) and sgRNAs were diluted to 3 $\mu$M with TE buffer (Synthego Inc.). The RNP complex was assembled by incubating Cas9 protein, sgRNAs and at a ratio of 1.3:1 for 10 min at room temperature. Transfection solution was prepared by incubating 3.5 $\mu$l Lipofectamine CRISPRMAX (Thermo Fisher Scientific) and 25 $\mu$l Opti-Mem per sample for 5 min at room temperature. RNP was added to the transfection solution and the mixture was incubated for 20 min at room temperature. Cells were trypsinized, washed with PBS, and counted. $10^5$ cells were used per transfection reaction. As a control, sgRNA was omitted from the RNP complex assembly. Cells were cultured for a total of 72 h, at which time the editing efficiency of the pooled cells was determined by PCR amplification of the appropriate genomic region, followed by DNA sequencing of the PCR product. Polyclonal pools were

identified at this point. Single-cell clones were isolated, cultured and validated, using genomic PCR, DNA sequencing, and Western blotting.

### Western blotting and co-immunoprecipitation assays (Co-IP)

In preparation for Western blotting and co-immunoprecipitation assays, cells were washed twice with PBS and harvested. Cell lysis was performed for 15 min on ice in an E1A buffer (20 mM NaH2PO4, 150 mM NaCl, 50 mM NaF, 0.5% [wt/vol] IGEPAL, 2.5 mM EDTA, 125 mM sodium pyruvate, and 10% [wt/vol] glycerol) with a proteinase inhibitor cocktail (1:1,000; MilliporeSigma). Lysates were centrifuged at 20,000 $g$ for 10 min at 4°C and protein concentrations were determined using a Pierce BCA Protein Assay Kit (Thermo Fisher Scientific) according to the manufacturer's instructions. For Western blots, inputs for Co-IPs, SDS loading buffer (250 mM Tris–HCl [pH 6.8], 8% [wt/vol] SDS, 0.2% [wt/vol] bromophenol blue, 40% [vol/vol] glycerol, 20% [vol/vol] $\beta$-mercaptoethanol) was added to the lysates, which were boiled for 7 min. For Co-IPs and IPs that were analyzed by LC/MS/MS, lysates were incubated with the appropriate antibody and an equivalent amount of the control IgG antibody, at 4°C overnight. Total amount of protein (per one IP) for DCP2 was 300 $\mu$g, DCP1a/1b 270 $\mu$g and for DDX6 300 $\mu$g. Protein A/G beads (Santa Cruz) were added to the lysates for 1 h at 4°C to capture the precipitate. The lysates were centrifuged for 5 min at ~1,000$g$. The beads were washed 3x with E1A buffer supplemented with a proteinase inhibitor cocktail. SDS loading buffer was added to lysates which were boiled for 7 min.

Lysates were separated by the SDS-polyacrylamide gel electrophoresis (SDS–PAGE) and electroblotted onto nitrocellulose membranes (BIO-RAD). Membranes were blocked for 30 min in 5% dry milk (AppliChem) reconstituted in TBS-T (20 mM Tris, 150 mM NaCl, 0.1% Tween [Millipore Sigma]) and incubated with the appropriate antibodies overnight at 4°C (DCP1a, DCP1b, DCP2, DDX6, EDC4, EDC3 1:1,000 in TBS-T, GAPDH 1:500 in TBS-T). The membranes were washed 3 × 10 min in TBS-T and incubated with the secondary anti-mouse or anti-rabbit antibodies for 1 h at room temperature. ECL Western blotting solution (Thermo Fisher Scientific) was used to visualize bands, and KwikQuant imager (Kindle Biosciences) was used to develop the images.

### LC/MS/MS analysis

SDS gels were stained with Colloidal Blue (Invitrogen). These stained regions were excised and reduced with TCEP; iodoacetamide was used for alkylation and, finally, digested with trypsin. Tryptic digests for DDX6 IP, DCP1a IP, and DCP1b IP were analyzed using the standard 90-min LC gradient on the Thermo Q Exactive Plus mass spectrometer. Tryptic digests for control, DCP1a KO, and DCP1b KO proteome cell analysis, were analyzed using an extended 4 h LC gradient on the Thermo Q Exactive Plus mass spectrometer. MS data were searched against the Swiss-Prot human proteome database (DDX6 IP: 6/4/2021, DCP1a IP, DCP1b IP, and proteomics study of control, DCP1a KO, and DCP1b KO cells: 10/10/2019) using MaxQuant (DDX6 IP: 1.6.17.0, DCP1a and DCP1b IP:1.6.3.3., proteomics study for control, DCP1a KO, and DCP1b KO cells: 1.6.15.0) Protein and

peptide false discovery rate was set to 1% and fold changes were calculated using intensity values.

For DDX6 IP, three replicas were performed and subjected to the analysis pipeline described above. Proteins were considered high confidence if they (1) had a minimum absolute fold change of 2, (2) had a q-value < 0.05, (3) were identified by a minimum of 2 razor + unique peptides in one of the sample groups compared, and (4) were detected in at least two of the replicates in one of the sample groups compared. Normalized intensity (as per DDX6 intensity) was used to generate heatmaps for the DDX6 IP (Fig 2B and C).

For the DCP1a and DCP1b IP, two replicas were performed, and proteins were considered high confidence if they (1) were identified by a minimum of 2 razor + unique peptides in the experimental condition, (2) were identified in both experimental replicas and (3) had minimum fold change of 2 (if not identified in the IgG control) or 20 (if the protein is also identified in the IgG control).

Proteins were considered high confidence in ctrl, DCP1a KO, and DCP1b KO cells if they (1) had a P-value < 0.05, (2) were identified by a minimum of 2 razor + unique peptides in any of the samples compared and (3) were detected in at least two of the triplicates in any of the groups compared.

iBAQ values were used to determine the relative amount of protein within the sample (42). To analyze DDX6 IP, background generated iBAQ was subtracted (IgG IP iBAQ was subtracted from DDX6 IP iBAQ), and the new iBAQ values (for DCP1a, DCP1b, EDC3, and EDC4) were scaled to DDX6 (DDX6 iBAQ was set to 1). Newly generated iBAQ values were used to generate a figure panel 2E. To perform the same analysis for DCP1a and DCP1b IPs in control cells, the background iBAQ values (IgG IP) was subtracted, the IP-ed protein iBAQ value was set to 1, and iBAQ value for the DCP1b and DCP1a was determined, respectively. Please see supplementary files with DDX6 IP, DCP1a IP, DCP1b IP, and proteomics study of control, DCP1a KO, and DCP1b KO datasets.

### SLAM-seq/GRAND-SLAM

A SLAM-seq Kinetic Kit was purchased from LEXOGEN and the manufacturer's instructions were followed. Briefly, cells were cultured to appropriate density and media containing 100 $\mu$M 4SU was added. We used parental line HCT116 for control, DCP1a KO cells (A3 single-cell clone) and DCP1b KO cells (B2 single-cell clone). For the pulse-chase, experiment cells were cultured for 24 h, whereas the fresh 4SU media was exchanged every 3 h and they were maintained in a light-free environment. At 24 h, the 4SU media was exchanged for 10x uridine-containing media and the first set of cells were collected. The cells were then collected again at 4, 8, and 24 h after the labeling was stopped.

For the progressive 4SU labeling experiment, the cells were cultured for 0 and 4 h in 100 $\mu$M 4SU media, when maintained in a light-free environment. For control, CRISPR control cell line was used (clone M17, parental cell line HCT116 was transfected with cas9 but not with guide RNAs), DCP1a KO cells (A3 clone) and DCP1b KO cells (B2 clone). Cells were lysed using TRIZOL and lysates were subsequently frozen at −80°C. All RNA isolation steps were performed in light-free environment. RNA concentration was determined using NanoDrop (Implen) and 5 $\mu$g of RNA was

alkylated by incubation at 50°C for 15 min, with a mixture of 100 mM iodoacetamide, 25 $\mu$l of organic solvent and 5 $\mu$l of sodium phosphate. After the reaction was terminated, light-free conditions were no longer used. Ethanol precipitation was performed to isolate the RNA according to manufacturer instructions (LEX-OGEN). mRNA libraries were prepared using the QuantSeq 3′ mRNA-Seq Library Prep Kit FWD (Illumina). Quality of mRNA libraries was determined using Agilent Tape Station and mRNA was sequenced at 75 bp single read sequencing using NextSeq 500 (Illumina).

### Dot blot analysis of 4SU labeled RNA

4SU incorporation was confirmed with dot blot analysis of 4SU labeled RNA, before alkylation. 4SU labeled RNA was biotinylated with EZ-link Iodoacetyl-LC-Biotin (Pierce), resulting in irreversible biotinylation as described (60). Briefly, 350 $\mu$l (50 $\mu$g of RNA diluted in nuclease-free H2O) of 4SU labeled RNA was incubated with 50 $\mu$l 10x Biotinylation Buffer (100 mM Tris pH 7.4, 10 mM EDTA in nuclease-free H$_2$O) and 100 $\mu$l of EZ-link Iodoacetyl-LC-Biotin (1 mg/ml in dimethylformamide) at room temperature for 1.5 h when rotating in light-free conditions. RNA was isolated by chloroform extraction and in 2 ml Phase Lock Gel heavy tubes to reduce the loss of RNA. RNA was precipitated with 5 M NaCl and isopropanol to the water phase, centrifuged, and the precipitate was washed with ethanol. Zeta membrane (Bio-Rad) was incubated in nuclease-free H$_2$O when rocking for 10 min and air-dried for 5 min. 5 $\mu$l of 200 ng/$\mu$l RNA dilution in ice-cold dot blot binding buffer (10 mM NaOH, 1 mM EDTA) was applied to the Zeta membrane by pipetting. Biotinlyated, non-4SU RNA was used as a control. The membrane was air-dried for 5 min, incubated for 30 min in 40 ml blocking buffer (20 ml 20% SDS with 20 ml 1x PBS pH 7–8, and EDTA to the final concentration of 1 mM) with rocking, and then incubated with 10 ml of 1:1,000 streptavidin-horseradish peroxidase for 15 min. The membrane was washed twice in 40 ml PBS + 10% SDS (20 ml PBS + 20 ml 20% SDS) for 5 min, twice in 40 ml PBS + 1% SDS (38 ml PBS + 2 ml 20% SDS) for 5 min, and twice in 40 ml PBS + 0.1% SDS (40 ml PBS + 200 $\mu$l 20% SDS) for 5 min. Residual buffer was removed by blotting and membrane-bound HRP was visualized as described above.

### Data analysis for SLAM-seq dataset

Slam-Dunk, in conjunction with the globally refined analysis of newly transcribed RNA and decay rates using SLAM-seq (GRAND-SLAM), and finally GrandR, was used to analyze SLAM-seq datasets (48, 49). Slam-Dunk is an automated SLAM-seq data analysis pipeline featuring a MultiQC plugin for diagnostic and quality assurance purposes (48, 49). Slam-Dunk output was used to run GRAND-SLAM. Slam-Dunk v0.4.3 was used and the following command was used to run this pipeline:

slamdunk all -r /mnt/e/slamseq_run_9_9_2021/hg19_no_alt_analy sis_set.fa -b /mnt/e/slamseq_run_9_9_2021/pure_UTR_3_hg19_ensem ble.bed -o /mnt/e/slamseq_run_9_9_2021//mnt/e/slamseq_ run_9_9_2021/sample_file.tsv

GRAND-SLAM's output includes the proportion and the corresponding posterior distribution of new and old RNA for each gene.

GRAND SLAM v2.0.6 version was used and the following command was used to run the pipeline:

gedi -e Slam -genomic homo_sapiens.75 -prefix test2/24 h -progress -plot -D -full -reads bams.bamlist

SLAM-DUNK can be found at https://t-neumann.github.io/slamdunk/ and GRAND-SLAM at https://github.com/erhard-lab/gedi/wiki/GRAND-SLAM.

The output of GRAND-SLAM was used as input for GrandR (50). GrandR was used to estimate half-lives and synthesis rates.

In the pulse-chase experiment, genes with 100 reads in all the samples are retained and normalized using DESeq estimated size factors. For the progressive labeling with 4SU experiment, genes with 200 reads in all the samples are retained and normalized using DESeq estimated size factors.

### Cell viability assays

To evaluate cell viability for pulse-chase assay, cells were cultured to an appropriate confluence and media containing 0 or 100 $\mu$M 4SU. Cells were cultured for 24 h, in the presence of 4SU when supplying fresh 4SU media every 4 h. At 24 h, the media was replaced with DMEM without 4SU, and 24 h later cell viability was evaluated.

To evaluate cell viability for progressive labeling assay, cells were cultured to an appropriate confluence and media containing 0, 50, 100, or 200 $\mu$M 4SU was added. Cells were cultured in the presence of 4SU for 6 h when supplying fresh 4SU media every 3 h. At 6 h the media was replaced with DMEM without 4SU and 24 h later cell viability was evaluated. Culture media was collected, and cells were harvested for evaluation of cell death.

To evaluate cell death, cells were collected by trypsinization (Corning), washed with PBS, and stained with either nothing or with propidium iodide (PI) and/or Annexin V. To stain for Annexin V, a BD Pharmingen PE Annexin V Apoptosis Detection Kit was used. Cell death was assessed with a CytoFLEX S Flow Cytometer (Beckman Coulter).

### Ontology analysis

WEB-Based Gene Set Analysis Toolkit (WebGestalt) is a functional enrichment analysis web tool that was used to examine the ontology of different sets of proteins or mRNAs (61). The over-representation analysis method was used, the reference set was genome (Fig 2F) or genome-coding (Fig 5D and E), top 10 (exception: top 15 for DCP1a and DCP1b IPs in Fig 3), and categories for Molecular Function, Biological Process and Cellular Component were combined and analyzed. Enrichment ratio accounts for the number of proteins/genes in the input list versus in the GO group. Only the relevant and statistically significant categories were shown in the bar charts, full lists are provided as supplemental data (add reference to specific figure).

### Cellular fractionation

Cellular fractionation assay was performed as previously described (62).

### Superose 6 gel filtration

Concentrated protein lysates were generated as described above, specifically, 2,000 $\mu$g (500 $\mu$l) of the protein per sample was used. The samples were resolved using a Superose 6 10/300 gel filtration column (Cytiva) equilibrated with SEC buffer (20 mM HEPES pH 7.5, 100 mM NaCl, 3 mM $\beta$-mercaptoethanol, 0.2 mM PMSF). The column was calibrated using Bio-Rad's Gel Filtration Standard (e.g., thyroglobulin, 670 kD; $\gamma$-globulin, 158 kD; ovalbumin, 44 kD; myoglobulin, 17 kD; vitamin B12, 1.35 kD). Column fractions were subjected to Western blots, as described above.

### Genomic PCR

Total DNA was isolated according to the manufacturer's instructions using Easy-DNA Kit (Invitrogen). 10 ng of gDNA was used for the reaction and 0.5 Taq DNA polymerase (Thermo Fisher Scientific) (per 50 $\mu$l total reaction volume) was used.

### Agarose DNA gel electrophoresis

Separation of DNA fragments was performed as previously described (63).

### Q-RT-PCR

Total RNA was isolated according to the manufacturer's instructions using TRIzol (Thermo Fisher Scientific). 200 ng of RNA was reverse transcribed into cDNA using the High Capacity cDNA Reverse Transcription Kit (Thermo Fisher Scientific). RT-PCR was performed with Fast SYBR Green Master Mix (Applied Biosystems). Gene values were normalized to GAPDH gene expression.

### Protein level quantification with image J

ImageJ software was used to quantify protein levels of DCP1a and DCP1b in Figs 1B and 3 (64). Image brightness was adjusted to remove the background. In the analysis options (subsection: gel), lanes were selected. The area for each lane is quantified. Protein of interest was normalized with the housekeeping protein (GAPDH), and the control condition was set to 1.

### Reagents

#### *Antibodies*
The following antibodies were used: DCP1a (ab47811; Abcam), DCP1b (mAb #13233; Cell Signaling Technologies), EDC4 (ab72408; Abcam), DCP2 (ab28658; Abcam), EDC3 (mAb #14495; Cell Signaling Technologies), GAPDH (mAb #5174; Cell Signaling Technologies), DDX6 (NB200-192; Novus Biologicals), and p53 (sc-6243 X; Santa Cruz Biotechnology).

#### *Enzymes*
SpCas9 2NLS Nuclease (no cat number; Synthego Inc.) was used.

### Kits

The following kits were used: Gene Knockout Kit v2—human DCP1A—1.5 nmol (no cat number; Synthego Inc.); Gene Knockout Kit v2—human DCP1B—1.5 nmol (no cat number; Synthego Inc.); SLAMseq Kinetics Kit—Anabolic Kinetics Module (061.24; Lexogen); SLAMseq Kinetics Kit—Catabolic Kinetics Module (062.24; Lexogen); QuantSeq 3' mRNA-Seq Library Prep Kit FWD for Illumina (015.24; Lexogen); Mycoplasma PCR Detection Kit (LookOut; Sigma-Aldrich); and Colloidal Blue Staining Kit (LC6025; Invitrogen).

### Non-standard chemicals

Plasmocin (ant-mpp; InvivoGen), Lipofectamine plus (CMAX0003; Invitrogen), and Zeta membrane (162-0153; Bio-Rad) were used.

### Biological resources

Various biological resources were used in the study: HEK293T, Human Cell Line, ATCC: CRL-3216; HCT116, Human Cell Line, ATCC: CCL-247; HCT116-derived cas9 control cell line (M17, M4); HCT116-derived DCP1a KO cell line (single-cell clone A3, polygenic clone A9); HCT116-derived DCP1b KO cell line (single-cell clone B2, single-cell clone B18); shLUC (SHC007; Sigma-Aldrich); psPAX2 (Addgene plasmid # 12260; Addgene); pMD2.G (Addgene plasmid # 12259; Addgene); Lentiviral plasmid shDCP1a (CAT#: TL305089; Origene); Lentiviral plasmid shDCP1b (CAT#: TL305088; Origene); and Lentiviral expression pDCP1b (CAT#: RC207398L1; Origene).

### Primers/CRISPR guide

Primers for validation of DCP1a KO clone (genomic PCR):
Forward: 5'-CTCGATGGCACTTTCCTGTCCGC-3',
Reverse: 5'-AATCCCAAGCGGACCCGACACC-3'.
Primers for validation of DCP1b KO clone (genomic PCR):
Forward: 5'-TCGCCGTGGGTTCTCGGTT-3'
Reverse: 5'-ACTTCCAGGGCGTCAACGTCTCC-3'.
Primers for validation of RE clones(genomic PCR):
Forward: 5'-CAATAGATCGCAGAACGAGCG-3'
Reverse: 5'-GGGGTCGAACACATCCAAGA-3'
Primers for DCP1b (qRT-PCR):
Forward: 5'-GCCACCACAGGCCTATTTCA-3'
Reverse: 5'-CATGAGCCTGATGTCCTACTGTCT-3'
Primers for BAX (gPCR):
Forward: 5'-TCCCCCCGAGAGGTCTTTT-3'
Reverse: 5'-CGGCCCCAGTTGAAGTTG-3'
CRISPR sgRNA guides for editing of the response element of p53 on the DCP1b promoter:
CCACUGGCAACGACAAGCCC.

### Data analysis/statistical analysis

The DDX6 IP had 3 replicas and was performed in control cells (HCT116), DCP1a KO (derived from HCT116), DCP1b KO (derived from HCT116). q-values and P-values were calculated. High-confidence targets were defined as significant as described above. Intensities were used with the R package *mixomics* to plot PCA analysis (65).

DCP1a IP and DCP1b IP each had two replicas and were performed in control cells only (HCT116). High-confidence targets were defined as significant as described above.

Proteomics LC/MS/MS study of control cells (HCT116), DCP1a KO cells, and DCP1b KO cells was performed in triplicates. P-value was calculated. High-confidence targets were defined as significant as described above. Intensities were used with the R package *mixomics* to plot PCA analysis (65).

The pulse-chase experiment has one replica. Genes with 100 reads in all the samples are retained and normalized using DESeq estimated size factors.

The progressive labeling with 4SU experiment has three replicas; genes with 200 reads in all the samples are retained and normalized using DESeq estimated size factors.

### Novel programs, software, algorithms

SLAM-seq/GRAND-SLAM programs are available as referenced. Code notebook is available on zenodo (https://zenodo.org/records/10725429).

### Websites/database referencing

To define protein domains, the web resource tool SMART 9.0 was used (http://smart.embl-heidelberg.de/). The normal mode was selected and after inputting the protein sequence of interest, outlier homologue, PFAM domains, signal peptides, and internal peptides were selected. This online tool was used to define the low-complexity regions in each protein. Other protein domains were sourced from recent publications (38).

To perform ontology analysis, the resource tool WebGestalt 2019 was used (https://www.webgestalt.org/). Network analysis or over-representation analysis was used. Network analysis settings were organism of interest: Homo sapiens, method of interest: network topology-based analysis, functional database: network (PPI BIOGRID), select gene ID type: gene symbol, upload gene list: uploaded as appropriate, network construction method: network expansion, set number of top-ranking neighbors: 10, significance level: top 10, highlight: seeds.

For over-representation analysis, the settings were organism of interest: Homo sapiens, method of interest: over-representation analysis, functional database: geneontology (select for Biological Process, Molecular Function, Cellular Localization separately), select gene ID type: gene symbol, upload gene list: uploaded as appropriate, select reference set: genome/genome protein-coding (as described above), minimum number of genes for a category: 5, maximum number of genes for a category: 2,000, multiple test adjustment: BH, significance level: top 10 or top 15 (as described above), number of categories expected from set cover: 10, number of categories visualized in the report: 40, color in DAG: continuous.

SLAM-seq QC analysis with MultiQC integrated into SlamDunk analysis was used to analyze our SLAM-seq dataset (https://t-neumann.github.io/slamdunk/). This analysis pipeline was coupled this with GRAND-SLAM analysis which is available at https://github.com/erhard-lab/gedi/wiki/GRAND-SLAM. GrandR (in R studio) was used to estimate half-lives and synthesis rates and is available from CRAN (50). GrandR version 0.2.2. was used.

To create Figs 1A and 4A https://www.biorender.com/ was used.

# Data Availability

The (SLAM-seq) pulse-chase experiment raw sequences are available through NCBI SRA (PRJNA1081121). The (SLAM-seq) progressive labeling experiment raw sequences are available through NCBI SRA (PRJNA1015671). Any dataset not available in the supplementary files can be requested from the corresponding authors.

# Supplementary Information

# Acknowledgements

We thank Andrew Kossenkov Ph.D. (Bioinformatics Facility, Wistar Institute), Hsin-Yao Tang Ph.D. (Proteomics Facility, Wistar Institute), Sonali Majumdar M.S. (Genomics Facility, Wistar Institute), Fadia Ibrahim Ph.D. (Thomas Jefferson University), Tobias Neumann, Ph.D., Florian Erhard Ph.D., Alison Moss, Ph.D., Yohei Kirino, PhD, Mike Kiledjian Ph.D., Alexander Mazo, Ph.D., Emad Alnemri, Ph.D., and Amanda Oran Ph.D.

## Author Contributions

I Vukovic: conceptualization, resources, data curation, software, formal analysis, validation, investigation, visualization, methodology, and writing—original draft, review, and editing.
SM Barnada: resources, software, visualization, methodology, and writing—review and editing.
JW Ruffin: resources, software, visualization, methodology, and writing—review and editing.
J Karlin: resources and investigation.
RK Lokareddy: resources, investigation, visualization, and methodology.
G Cingolani: supervision, methodology, and writing—review and editing.
SB McMahon: conceptualization, resources, data curation, formal analysis, supervision, funding acquisition, validation, investigation, visualization, methodology, project administration, and writing—original draft, review, and editing.

## Conflict of Interest Statement

The authors declare that they have no conflict of interest.

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
