## [Reviewer comments · Life Science Alliance]

Life Science Alliance

Non-redundant roles for the human mRNA decapping cofactor paralogs DCP1a and DCP1b

Ivana Vukovic, Samantha Barnada, Jonathan Ruffin, Jonathan Karlin, Ravi Lokareddy, Gino Cingolani, and Steven McMahon
DOI: 10.26508/lsa.202402938

Corresponding author(s): Steven McMahon, Thomas Jefferson University and Ivana Vukovic, Thomas Jefferson University

Review Timeline:

Submission Date:	2024-07-11
Editorial Decision:	2024-07-30
Revision Received:	2024-08-21
Accepted:	2024-08-21

Transaction Report:

Please note that the manuscript was reviewed at Review Commons and these reports were taken into account in the decision-making process at Life Science Alliance.

Manuscript number: RC-2023-02218R

Corresponding author(s): Steven, McMahon

1. General Statements [optional]

We were pleased to receive the encouraging critiques and very much appreciate the Reviewer's specific comments and suggestions. In this revised version of our manuscript, we have made a number of substantive additions and modifications in response to these comments/suggestions. We hope you agree that the study is now improved to the point where it is suitable for publication.

Reviewer #1 (Evidence, reproducibility and clarity (Required)):

Summary

This study describes efforts to characterize differences in the roles of the two related human decapping factors Dcp1a and Dcp1b by assessing mRNA decay and protein associations in knockdown and knockout cell lines. The authors conclude that these proteins are non-redundant based on the observations that loss of DCP1a versus Dcp1b impacts the decapping complex (interactome) and the transcriptome differentially.

Major comments

- While the experiments appear to be well designed and executed and the data of generally high quality, the conclusions are drawn without sufficient consideration for the fact that these two proteins form a heterotrimeric complex. The authors assume that there are distinct homotrimeric complexes rather than a single complex with both proteins in. Homotrimers may have new/different functions not normally seen when both proteins are expressed. Thus while it is acceptable to infer that the functions of these two proteins within the decapping complex are distinct, it is not clear that they act separately, or that complexes naturally exist without one or the other. A careful evaluation of the relative ratios of Dcp1a and b overall and in decapping complexes would be informative if the authors want to make stronger statements about the roles of these two factors.

RESPONSE: Thank you for this valuable comment. We have substantially edited the manuscript to incorporate these points. Examples include a detailed analysis of iBAQ values for the DDX6, DCP1a, and DCP1b interactomes (which now allows us to estimate the ratios of DCP1a and DCP1b in these complexes) and cellular fractionation to interrogate complex integrity (using Superose 6).

- The concept of buffering is not adequately introduced and the interpretation of observations that RNAs with increased half life do not show increased protein abundance - that Dcp1a/b are

involved in transcript buffering is nebulous. In order to support this interpretation, the mRNA abundances (NOT protein abundances) should be assessed, and even then, there is no way to rule out indirect effects.

RESPONSE: Thank you for this comment. In the revised version of the manuscript, we introduced the concept of transcript buffering at an earlier stage as one of the potential explanations for our findings. We were also able to use a new algorithm (grandR) to estimate half-lives and synthesis rates from our data. These new data add strength to the argument that DCP1a and DCP1b are linked to transcript buffering pathways.

- It might be interesting to see what happens when both factors are depleted to get an idea of the overall importance of each one.

RESPONSE: In our work we tried to emphasize the differences between the two paralogs. We believe that doing double knockout or knockdown would mask the distinct impacts of the paralogs. In data not included in this study, we have shown that cells lacking both DCP1a and DCP1b are viable. We did check PARP cleavage in the CRISPR generated cell pools of DCP1a KO, DCP1b KO, and the double KO. The WB measuring the PARP cleavage is shown in the supplemental material (Supplementary Material: Replicates)

- The algorithms etc used for data analysis should be included at the time of publication. Version number and settings used for SMART to define protein domains, and webgestalt should be indicated

RESPONSE: We apologize for this oversight. Version number and settings used for the webtools (SMART, Webgestalt) are now included. The analysis pipeline for half-lives and synthesis rates estimation as well as all the files and the code needed to generate the figures in the paper are available on zenodo (<https://zenodo.org/records/10725429>).

- Statistical analysis is not provided for the IP experiments, the number of replicates performed is not indicated and quantification of KD efficiency are not provided.

RESPONSE: The number of replicates performed in each experiment is now clearly indicated and quantifications of knockdown efficiency are provided (Supplemental Figure 3A and 3B, Figure 3A, Figure 3B).

- The possibility that the IP Antibody interferes with protein-protein interactions is not mentioned.

RESPONSE: Thank you for this comment. The revised manuscript includes a discussion of the antibody epitope location and the potential for impact on protein-protein interactions.

Minor comments

- P4 - "This translational repression of mRNA associated with decapping can be reversed, providing another point at which gene expression can be regulated (21)" - implies that decapping can be reversed or that decapped RNAs are translated. I don't think this is technically true.

RESPONSE: There have been several studies that document the reversal of decapping. These findings are summarized in the following reviews.

Schoenberg, D. R., & Maquat, L. E. (2009). Re-capping the message. Trends in biochemical sciences, 34(9), 435-442.

Trotman, J. B., & Schoenberg, D. R. (2019). A recap of RNA recapping. Wiley Interdisciplinary Reviews: RNA, 10(1), e1504.

- P11 - how common is it for higher eukaryotes to have 2 DCP genes?

RESPONSE: Metazoans have 2 DCP1 genes.

- Fig S1 - says "mammalian tissues" in the text but the data is all human. The statement that "expression analyses revealed that DCP1a and DCP1b have concordant rather than reciprocal expression patterns across different mammalian tissues (Supplemental Figure 1)" is a bit misleading as no evidence for correlation or anti-correlation is provided. Also co-expression is not strong support for the idea that these genes have non-redundant functions. Both genes are just expressed in all tissues - there's no evidence provided that they are concordantly expressed. In bone marrow it may be worth noting that one is high and the other low - i.e. reciprocal.

RESPONSE: We appreciate this comment. We have corrected the interpretation of the aforementioned dataset. We have also incorporated a more detailed discussion in the text of the paper. As the Reviewer pointed out, there are a subset of tissues where their expression appears to be reciprocal.

- Fig 1A - it is not clear what the different colors mean. Does Sc DCP1 have 1 larger EVH or 2 distinct ones. Are the low complexity regions in Sc DCP2 the SLiMs.

RESPONSE: Thank you for this comment. We have corrected this ambiguity to reflect that Sc DCP1 has one EVH1 domain that is interconnected by a flexible hinge. The low-complexity regions typically contain short linear motifs (SLiMs), however, not all low-complexity regions have been verified to contain them. In the figure, only low-complexity regions are shown. The text of the paper refers only to verified SLiMs .

- P11 - why were HCT116 cells selected?

RESPONSE: HCT116 cells are an easily transfectable human cell line and have been widely used in biochemical and molecular studies, including studies of mRNA decapping (see references below). Since decapping is impacted by viral proteins we avoided the use of other commonly used cell models such as HEK293T or HeLa.

<https://pubmed.ncbi.nlm.nih.gov/?term=decapping+hct116&sort=date&size=200>

- Fig 1B - what are the asterisks by the RNA names? Might be worth noting that over-expression of DCP1b reduced IP of DCP1a. There's no quantification and no indication of the number of

times this experiment was repeated. Data from replicates and quantification of the knockdown efficiency in each replicate would be nice to see.

RESPONSE: Thank you for this comment. Asterisks indicate that those bands were from a second gel, as DCP1a and DCP1b run at approximately the same molecular weight. We have now included a note in our figure legend to indicate this. The knockdown efficiency is provided (Figure 3 and Supplemental Figure 3). We also noted the number of replicas for each IP in figure 1. The replicas are provided as supplementary material (Supplementary Materials: Replicates).

• Fig 1C/1D - why are there 3 bands in the DCP1a blot? Quantification of the IP bands is necessary to say whether there is an effect or not of over-expression/KO.

RESPONSE: The additional bands in DCP1a blots are background. When we stained the whole blot for DCP1a, in cells which with complete DCP1a KO cells (clone A3), these bands still appear (Supplementary Material: Validation of the KO clones). Quantifications of the bands in the overexpression experiments is now provided.

• Fig 3 - is it possible that differences are due to epitope positions for the antibodies used for IP?

RESPONSE: We do not believe so. DCP1a antibody binds roughly 300-400 residues on DCP1a, and DCP1b antibody binds around Val202. Antibodies therefore do not bind DCP1a or DCP1b low-complexity regions (which are largely responsible for interacting with the decapping complex interactome). Antibodies don't bind the EVH1 domains or the trimerization domain, which are needed for their interaction with DCP2 and each other.

• Fig 5A - the legend doesn't match the colors in the figure. It is not clear how the $p < 0.05$ high confident genes are identified - only some of the genes with $p < 0.05$ are colored red.

RESPONSE: Thank you for this comment. We have corrected this issue in the revised version of the paper. High-confidence proteins are those with $p < 0.05$, and have been identified by a minimum of 2 razor + unique peptides in any of the samples compared and detected in at least 2 of the triplicates in any of the groups compared. Thus, the proteins on the volcano plot can have a $p < 0.05$ but if they don't fulfill other requirements they will not be labeled with a color (to signal high-confidence proteins). We understand that this is not a typical use of the volcano plots. The proteins are now labeled as "high-confidence" or "NS" on the plot instead.

• Fig 5E and F - x-axis should be \log_2 fold change

RESPONSE: Thank you for this comment. We have corrected this issue in the revised version of the paper.

• There are a few more recent studies on buffering that should be cited and more discussion of this in the introduction is necessary if conclusions are going to be drawn about buffering.

RESPONSE: We have included a discussion of transcript buffering in the introduction.

• The heatmaps in figure 2 are hard to interpret.

Full Revision

RESPONSE: To clarify the heatmaps, we included a more detailed description in the figure legends, have enlarged the heatmaps themselves, and have added more extensive labeling.

Reviewer #1 (Significance (Required)):

- Strengths: The experiments appear to be done well and the datasets should be useful for the field.
- Limitations: The results are overinterpreted - different genes are affected by knocking down one or other of these two similar proteins but this does not really tell us all that much about how the two proteins are functioning in a cell where both are expressed.
- Audience: This study will appeal most to a specialized audience consisting of those interested in the basic mechanisms of mRNA decay. Others may find the dataset useful.
- This study might complement and/or be informed by another recent study in BioRxiv - <https://doi.org/10.1101/2023.09.04.556219>
- My field of expertise is mRNA decay - I am qualified to evaluate the findings within the context of this field. I do not have much experience of LC-MS-MS and therefore cannot evaluate the methods/analysis of this part of the study.

Reviewer #2 (Evidence, reproducibility and clarity (Required)):

The authors provide evidence that Dcp1a and Dcp1b - two paralogous proteins of the mRNA decapping complex - may have divergent functions in a cancer cell line. In the first part, the authors show that interaction of Dcp2 with EDC4 is diminished upon depletion of Dcp1a but not affected by depletion of Dcp1b. The results have been controlled by overexpression of Dcp1b as it may be limiting factor (i.e. expression levels too low to compensate for depletion of Dcp1a reduced interaction with EDC3/4 while depletion of Dcp1b lead to opposite and increase interactions). They then defined the protein interactome of DDX6 in parental and Dcp1a or Dcp1b depleted cells. Here, the authors show some differential association with EDC4 again, which is along results shown in the first part. The authors further performed SLAM-seq and identified subsets of mRNA whose decay rates are common but also different upon depletion with Dcp1a and Dcp1b. Interestingly, it seems that Dcp1a preferentially targets mRNAs for proteins regulating lymphocyte differentiation. To further test whether changes in RNA decay rates are also reflected at the protein levels, they finally performed an MS analysis with Dcp1a/b depleted cells. However no significant overlap with mRNAs showing altered stability could be observed; and the authors suggested that the lack of congruence reflects translational repression.

Major comments:

1. While functional difference between Dcp1a and Dcp1b are interesting and likely true, there

are overinterpretations that need correction or further evidence for support.

Sentences like "DCP1a regulates RNA cap binding proteins association with the decapping complex and DCP1b controls translational initiation factors interactions (Figure 2E)" sound misleading. While differential association with proteins has been recognised with MS-data, it does not necessary implement an active process of control/regulation. To make the claim on 'control/regulation', and inducible system or introduction of mutants would be required.

RESPONSE: This set of comments were particularly useful in helping us refine the presentation of our findings. We have edited our manuscript to be more specific about the limits of our data.

2. The MS analysis is not clearly described in the text and it is unclear how authors selected high-confident proteins. The reader needs to consider the supplemental tables to find out what controls were used. Furthermore, the authors should show correlation plots of MS data between replicates. For instance, there seems to be limited correlation among some of the replicates (e.g. Dcp1b_ko3 sample, Fig. 2c). Any explanation in this variance?

RESPONSE: We have now included a clear description of how all high-confidence proteins were selected in the Methods and Results sections. The revised manuscript also includes a more thorough description of the controls used and the number of replicates for individual experiments. The PCA plots have now been included where appropriate. The variance in this sample is likely technical.

3. GO analysis for the proteome analysis should consider the proteome and not the genome as the background. The authors should also indicate the corrected P-values (multiple testing) FDRs.

RESPONSE: Webgestalt uses a reference set of IDs to recognize the input IDs, and it does not use it for the background analysis in the classical sense. We repeated a subset of our proteome analyses using the 'genome-protein coding' as background and obtained the same result as in our original analysis. All ontology analyses now include raw p-values and/or FDRs when appropriate.

4. Fig 2E. The figures display GO enrichments needs better explanation and additional data can be added. The enrichment ratio is not explained (is this normalised?) and p-values and FDRs, number of proteins in respective GO category should be added.

RESPONSE: More thorough explanations of the GO enrichments are now included. The supplemental data contains all p-values (raw and adjusted), as well as the number of proteins in each GO category. The Enrichment ratio is normalized and contains information about the number of proteins that are redundant in multiple groups. GO Ontology analyses are now displayed with p-values and/or FDR values, and in this case the enrichment ratio contains information regarding the number of proteins found in our input set and the number of expected proteins in the GO group. The network analysis shows the FDR values and the number of proteins found in the groups compared.

Minor:

5. These studies were performed in a colorectal carcinoma cell line (HCT116). The authors

should justify the choice of this specialised cell line. Furthermore, one wonders whether similar conclusions can be drawn with other cell lines or whether findings are specific to this cancer line.

RESPONSE: The study that is currently in pre-print in BioRxiv (<https://doi.org/10.1101/2023.09.04.556219>) utilized HEK293Ts and found similar results to ours when examining the various relationships between the core decapping core members.

6. Fig. 1B. It is unclear what DCP1b* refers to? There are bands of different size that are not mentioned by the authors - are those protein isoforms or what are those referring to? A molecular marker should be added to each Blots. Uncropped Western images and markers should be provided in the Supplement.

RESPONSE: The asterisk indicates that these images came from a second western blot gel (DCP1a and DCP1b have a similar molecular weight and cannot be probed on the same membrane). Uncropped western blot images and markers (as available) are provided in the supplement.

7. MS data submitted to public repository with access. No. indicated in the manuscript.

RESPONSE: MS data is submitted as supplementary datasets to the paper. It contains the analyzed data as well as the LCMSMS output. We are in the process of submitting the raw LSMSMS data to a public repository.

8. Fig 3. A Venn Diagram displaying the overlap of identified proteins should be added. GO analysis should be done considering the proteome as background (as mentioned above).

RESPONSE: A Venn diagram showing the overlap among the proteins identified is now included in the revised version.

Reviewer #2 (Significance (Required)):

Overall, this is a large-scale integrative -omics study that suggest functional difference between Dcp1 paralogues. While it seems clear that both paralogous have some different functions and impact, there are overinterpretations in place and further evidence would to be provided to substantiate conclusions made in the paper. For instance, while the interactions with Dcp2/Ddx6 in the absence of Dcp1a,b with EDC4/3 may be altered (Fig. 1, 2), the functional implications of this changed associations remains unresolved and not further discussed. As such, it remains somehow disconnected with the following experiments and compromises the flow of the study. The observed differences in decay-rates for distinct functionally related sets of mRNAs is interesting; however, it remains unclear whether those are direct or rather indirect effects. This is further obscured by the absence of any correlation to changes in protein levels, which the authors interpreted as 'transcriptional buffering'. In this regard, it is puzzling how the authors can make a statement about transcriptional buffering? While this may be an interesting aspect and concept of the discussion, there is no primary data showing such a functional impact.

As such, the study is interesting as it claims functional differences between DCP1a/b paralogous in a cancer cell line. Nevertheless, I am not sure how trustful the MS analysis and decay measurements are as there is not further validation. It would be interesting if the authors could go a bit further and draw some hypothesis how the selectivity could be achieved i.e interaction with RNA-binding proteins that may add some specificity towards the target RNAs for differential decay. As such, the study remains unfortunately rather descriptive without further functional insight.

Reviewer #3 (Evidence, reproducibility and clarity (Required)):

Review on "Non-redundant roles for the human mRNA decapping cofactor paralogs DCP1a and DCP1b" by Steven McMahon and co-workers

mRNA decay is a critical step in the regulation of gene expression. In eukaryotes, mRNA turnover typically begins with the removal of the poly(A) tail, followed by either removal of the 5' cap structure or exonucleolytic 3'-5' decay catalyzed by the exosome. The decapping enzyme DCP2 forms a complex with its co-activator DCP1, which enhances decapping activity. Mammals are equipped with two DCP1 paralogs, namely DCP1a and DCP1b. Metazoans' decapping complexes feature additional components, such as enhancer of decapping 4 (EDC4), which supports the interaction between DCP1 and DCP2, thereby amplifying the efficiency of decapping.

This work focuses on DCP1a and DCP1b and investigates their distinct functions. Using DCP1a- and DCP1b-specific knockdowns as well as K.O. cell lines, the authors find surprising differences between the DCP1 paralogs. While DCP1a is essential for the assembly of EDC4-containing decapping complexes and interactions with mRNA cap binding proteins, DCP1b mediates interactions with the translational machinery. Furthermore, DCP1a and DCP1b target different mRNAs for degradation, indicating that they execute non-overlapping functions.

The findings reported here expand our understanding of mRNA decapping in human cells, shedding light on the unique contributions of DCP1a and DCP1b to mRNA metabolism.

The manuscript tackles an interesting subject. Historically, the emphasis has been on studying DCP1a, while DCP1b has been deemed a functionally redundant homolog of DCP1a.

Therefore, it is commendable that the authors have taken on this topic and, with the help of knockout cell lines, aimed to dissect the function of DCP1a and DCP1b.

Despite recognizing the significance of the subject and approach, the manuscript falls short of persuading me. Following a promising start in Figure 1 (which still has room for improvement), there is a distinct decline in overall quality, with only relatively standard analyses being conducted. However, I do not want to give the authors a detailed advice on maximizing the potential of their data and presenting it convincingly. So, here are just a few key points for improvement:

Figure 1C: Upon closer examination, a faint band is still visible at the size of DCP1a in the DCP1a knockout cells. Could this be leaky expression of DCP1a? The authors should provide an in-depth characterization of their cells (possibly as supplementary material), including

identification of genomic changes (e.g. by sequencing of the locus) and Western blots with longer exposure, etc.

RESPONSE: Thank you for this comment. The in-depth characterization of our cells is now included in the Supplementary Material. DCP1a KO cells and DCP1b KO cells indicated as single cell clones have been confirmed to have no DCP1a or DCP1b expression. In Figure 1D and Figure 3, polyclonal pool cells were used as indicated (only for DCP1a KO).

Figure 2: It is great to see that the effects of the KOs are also visible in the DDX6 immunoprecipitation. However, I wonder if the IP clearly confirms that the KO cells indeed do not express DCP1a or DCP1b. In the heatmap in Figure 2B, it appears as if the proteins are only reduced by a log₂-fold change of approximately 1.5? Additionally, Figure 2 shows a problem that persists in the subsequent figures. The visual presentation is not particularly appealing, and essential details, such as the scale of the heatmap in 2B (is it log₂ fold?), are lacking.

RESPONSE: The in-depth characterization of our cells is included in the Supplementary Materials and confirms the presence of single-cell clones where indicated. As noted above, only Figure 1D and Figure 3 used DCP1a KO pooled cells. The heatmap in Figure 2B is scaled by row using the pheatmap function in R studio. The actual data for the heatmap comes from protein intensities from the LC-MS/MS analysis. We have improved the visual presentation in the revised manuscript.

Figure 3: I wonder why there are no primary data shown here, only processed GO analyses. Wouldn't one expect that DCP2 interacts mainly with DCP1a, but less with DCP1b? Is this visible in the data? Moreover, such analyses are rather uninformative (as reflected in the GO terms themselves, for instance, "oxoglutarate dehydrogenase complex" doesn't provide much meaningful insight). The authors should rather try to derive functional and mechanistic insights from their data.

RESPONSE: We have now revised this Figure to include primary data as well as the IP of DCP1a in DCP1b KO cells (single cell clones) and the IP of DCP1b in DCP1a KO cells (pooled cells). We identified EDC3 in the high-confidence protein pool. The EDC3:DCP1a interaction is enhanced in DCP1b KO cells. We also found that the EDC3:DCP1b interaction is less abundant in DCP1a KO cells. This is consistent with our data in Figures 1 and 2. DCP2 was not identified in the interactomes of either DCP1a or DCP1b. This is not unusual as DCP2 is highly flexible and the association between DCP1s with DCP2 is transient and facilitated by other proteins.

In Fig. 4 the potential of the approach is not fully exploited. Firstly, I would advocate for omitting the GO analyses, as, in my opinion, they offer little insight. Again, crucial information is missing to assess the results. While 75 nt reads are mentioned in the methods, the sequencing depth remains unspecified. Figure 4b should be included in the supplements. Furthermore, I strongly recommend concentrating on insights into the mechanisms of DCP1a and DCP1b-containing complexes. E.g. what characteristics distinguish DCP1a and DCP1b-dependent mRNAs? Are these targets inherently unstable? Why are they degraded? Are they known decapping substrates?

Full Revision

RESPONSE: Thank you for this comment. We have now revised this figure and have included information about sequencing depth and other pertinent information. We have been able to use a newly available algorithm (grandR) and were able to estimate half-lives and synthesis rates. This is a significant addition to the paper. We were also able to compare significantly impacted mRNAs (by DCP1a or DCP1b loss) to the established DCP2 target list.

In general, I suggest the authors revise the manuscript with a focus on the potential readers. Reduce Gene Ontology (GO) analyses and heatmaps, and instead, incorporate more analyses regarding the molecular processes associated with the different decapping complexes.

RESPONSE: We removed selected GO analyses and heatmaps from the main body of the manuscript (included as Supplementary Figures instead). For our LC-MS/MS datasets, we added iBAQ analyses of the DDX6 IP, DCP1a IP, and DCP1b IP in the control conditions. Cellular fractionation studies (using Superose 6 chromatography) were also added to the paper and allow us to interrogate decapping complex composition in more detail. The revised version of the manuscript includes a new 4SU labeling experiment (pulse-chase) as well as estimation of half-lives and synthesis rates in our conditions. Also included is relevant information about DCP1b transcriptional regulation.

Reviewer #3 (Significance (Required)):

The manuscript in its current form could benefit from substantial revisions for it to be considered impactful for researchers in the field.

July 30, 2024

RE: Life Science Alliance Manuscript #LSA-2024-02938

Steven B McMahon
Thomas Jefferson University
Cancer Biology
233 S. 10th Street
Philadelphia, PA 19104

Dear Dr. McMahon,

Thank you for submitting your revised manuscript entitled "Non-redundant roles for the human mRNA decapping cofactor paralogs DCP1a and DCP1b". We would be happy to publish your paper in Life Science Alliance pending final revisions necessary to meet our formatting guidelines.

- please be sure that the authorship listing and order is correct
- please upload your main manuscript text as an editable doc file
- please upload all figure files as individual ones, including the supplementary figure files; all figure legends should only appear in the main manuscript file after the references sections
- please add a Running Title and a Summary Blurb/Alternate Abstract to our system
- please add a Category for your manuscript in our system
- please add the Twitter handle of your host institute/organization as well as your own or/and one of the authors in our system
- please make sure the author order in your manuscript and our system match and that all authors are entered into the system
- please consult our manuscript preparation guidelines <https://www.life-science-alliance.org/manuscript-prep> and make sure your manuscript sections are in the correct order
- please add an Author Contributions section to your main manuscript text and the system
- please add a Conflict of Interest statement to your main manuscript text
- please provide one figure per file
- please add your primary and supplementary figure legends to the main manuscript text after the references section
- please remove figures from the manuscript text and leave them uploaded separately
- we encourage you to revise the figure legend for Figure 3 such that the figure panels are introduced in alphabetical order
- there is a callout for Figure 1E on pg. 6, and this figure doesn't have this panel... please correct
- there are callouts for Fig. 6D and E on pg. 8, and this figure hasn't been provided...please correct
- please add callouts for Figures 3F; S3A-B; and S5B-D to your main manuscript text

FIGURE CHECKS:

- please add sizes next to all blots
- files labeled ddx6 IP, dcp1a/b IPs, KO proteomics, pulse-chase analysis should be uploaded as Source Data files
- the 2 Supplemental Material files should be uploaded as Supplemental Figures

A. FINAL FILES:

B. MANUSCRIPT ORGANIZATION AND FORMATTING:

Sincerely,

Reviewer #3 (Comments to the Authors (Required)):

Considering the differing directions of the reviewers' comments, the authors have done a good job to bring the manuscript to a publishable state. I have no further objections and support its acceptance in LSA.

August 21, 2024

RE: Life Science Alliance Manuscript #LSA-2024-02938R

Dr. Steven B McMahon
Thomas Jefferson University
Cancer Biology
233 S. 10th Street
Philadelphia, PA 19104

Dear Dr. McMahon,

Thank you for submitting your Research Article entitled "Non-redundant roles for the human mRNA decapping cofactor paralogs DCP1a and DCP1b". It is a pleasure to let you know that your manuscript is now accepted for publication in Life Science Alliance. Congratulations on this interesting work.

DISTRIBUTION OF MATERIALS:

Again, congratulations on a very nice paper. I hope you found the review process to be constructive and are pleased with how the manuscript was handled editorially. We look forward to future exciting submissions from your lab.

Sincerely,
